# *In vitro* spermatogenesis in isolated seminiferous tubules of immature mice

Xuemin Feng[1☯], Takafumi Matsumura[1,2☯], Yuki Yamashita[2☯], Takuya Sato[1,2], Kiyoshi Hashimoto[3], Hisakazu Odaka[3], Yoshinori Makino[4], Yuki Okada[4], Hiroko Nakamura[5], Hiroshi Kimura[5], Teruo Fujii[6], Takehiko Ogawa[1,2]*

**1** Laboratory of Biopharmaceutical and Regenerative Sciences, Institute of Molecular Medicine and Life Science, Yokohama City University Association of Medical Science, Yokohama, Kanagawa, Japan, **2** Department of Regenerative Medicine, Yokohama City University Graduate School of Medicine, Yokohama, Japan, **3** Department of Urology, Yokohama City University School of Medicine, Yokohama, Kanagawa, Japan, **4** Laboratory of Pathology and Development, Institute of Molecular and Cellular Biosciences, The University of Tokyo, Tokyo, Japan, **5** Department of Mechanical Engineering, School of Engineering, Tokai University, Hiratsuka, Kanagawa, Japan, **6** Institute of Industrial Science, University of Tokyo, Meguro, Tokyo, Japan

☯ These authors contributed equally to this work.
* ogawa@yokohama-cu.ac.jp

**Data Availability Statement:** All relevant data are within the paper and its Supporting Information files.

**Funding:** This work was supported by a Grant-in-Aid for Scientific Research on the Innovative Area

## Abstract

Mouse spermatogenesis, from spermatogonial stem cell proliferation to sperm formation, can be reproduced *in vitro* by culturing testis tissue masses of neonatal mice. However, it remains to be determined whether this method is also applicable when testis tissues are further divided into tiny fragments, such as segments of the seminiferous tubule (ST), a minimal anatomical unit for spermatogenesis. In this study, we investigated this issue using the testis of an *Acrosin*-GFP/Histone H3.3-mCherry (*Acr*/H3) double-transgenic mouse and monitored the expression of GFP and mCherry as indicators of spermatogenic progression. Initially, we noticed that the cut and isolated stretches of ST shrunk rapidly and conglomerated. We therefore maintained the isolation of STs in two ways: segmental isolation without truncation or embedding in soft agarose. In both cases, GFP expression was observed by fluorescence microscopy. By whole-mount immunochemical staining, meiotic spermatocytes and round and elongating spermatids were identified as Sycp3-, crescent-form GFP-, and mCherry-positive cells, respectively. Although the efficiency was significantly lower than that with tissue mass culture, we clearly showed that spermatogenesis can be induced up to the elongating spermatid stage even when the STs were cut into short segments and cultured in isolation. In addition, we demonstrated that lowered oxygen tension was favorable for spermatogenesis both for meiotic progression and for producing elongating spermatids in isolated STs. Culturing isolated STs rather than tissue masses is advantageous for explicitly assessing the various environmental parameters that influence the progression of spermatogenesis.

"Ensuring integrity in gametogenesis" (18H05546), CREST by JST (JPMJCR21N1), a Grant for Strategic Research Promotion of Yokohama City University (SK2811) to T.O., and a KAKENHI grant (JP19J01276) from Japan Society for the Promotion of Science (JSPS) to T.M. The funders had no role in study design, data collection and analysis, decision to publish, or preparation of the manuscript.

**Competing interests:** The authors have declared that no competing interests exist.

## Introduction

Most organs can be divided into smaller anatomical subunits that can be regarded as a minimal functional unit. These functional units derived from particular organs can be maintained in culture as explants for a certain period of time. Organoids, which are self-organizing systems of stem cells and their progeny cells, are in many cases composed of such functional units, providing a useful tool for biomedical research [1].

Spermatogenesis takes place in the seminiferous tubules (STs), which are anatomical structures consisting of Sertoli cells on the inside, peritubular myoid cells on the outside, and basal lamina between them. Several different types of cells outside STs, collectively called interstitial cells, also play important roles in controlling spermatogenesis [2]. Approximately 11 STs are packed inside a testis of an adult mouse, and the individual STs have an average length of 140 mm [3]. Therefore, the functional minimum unit of the testis is presumed to be an ST segment of certain length with germ cells inside and interstitial cells around.

In 2011, we developed a culture system that reproduced the complete process of spermatogenesis, from spermatogonial stem cell proliferation to sperm formation, in an explanted mouse testis tissue. Offspring were obtained by micro-insemination with the haploid cells produced in the explants [4]. The explanted testis tissue pieces were about 1 mm$^3$ or larger in size and consisted of numbers of the functional units. In experimental systems, manipulating a single or a small number of functional units could serve to clarify the mechanistic details of physiological phenomena and pathological disorders. Namely, interactions between functional units as well as between a unit and the cells surrounding it could be examined experimentally. However, it has proven quite difficult to dissect the functional unit of spermatogenesis. In fact, only a few studies have reported the use of isolated segments of STs to promote or maintain spermatogenesis in culture. Some of these studies, using the testis of adult rats of 60 to 120 days of age, demonstrated the progression of a limited span of spermatogenesis lasting for several days [5–7]. Another recent study showed the progression of spermatogenesis for 40 days using a culture of STs in soft agar [8]. On the other hand, other investigations have tried to reconstruct testis tissue architecture from singly isolated testicular cells. These studies showed different extents of successful spermatogenesis [9–11]. Based on these previous studies and our tissue mass culture experience, we expected that spermatogenesis in an isolated ST would be possible.

However, our early attempts were hampered by the stubborn nature of the STs themselves, which contracted and shrunk rapidly upon culturing. We overcame this problem using two methods and demonstrated that the isolated STs could support spermatogenesis up to the stage of elongating spermatids. This result showed experimentally for the first time that a limited ST segment can be a functional unit for spermatogenesis. In addition, using this method, we demonstrated that spermatogenesis in the isolated STs responded sensitively to oxygen concentration. This was not fully demonstrated in previous studies culturing tissue masses. The ST culture method, therefore, is useful for investigating various factors and conditions that influence the progression and maintenance of spermatogenesis.

## Materials and methods

### Animals

*Acrosin (Acr)*-GFP transgenic mice [12, 13] (genetic background: ICR, C57BL/6, and their mixture) were provided by RIKEN BRC through the National Bio-Resource Project of MEXT, Japan. Histone H3.3-mCherry (H3) transgenic mice [14] were provided by Makino & Okada. *Acr*-GFP/Histone H3.3.3-mCherry double (Acr/H3) transgenic mice were generated by

crossing an *Acr*-GFP homogeneous female mouse with an H3 homogeneous male mouse. Mice aged 1–7 days postpartum (dpp) were euthanized by an anesthetic overdose using a medetomidine/midazolam/butorphanol (0.3/4/5) cocktail. Their testes were then collected. Mice were housed in specific pathogen-free, air-conditioned rooms at 24±1˚C and 55±5%, with a 13-h light/11-h dark lighting cycle. They were fed *ad libitum* with commercially available hard pellets (MF; Oriental Yeast). Drinking water was acidified to pH 2.8–3.0 by HCl. All animal experiments were performed in accordance with the Guidelines for Proper Conduct of Animal Experiments (Science Council of Japan) and were approved by the Institutional Committee of Laboratory Animal Experimentation (Animal Research Center of Yokohama City University; protocol no. F-A-20-038).

## Culture media and reagents

The culture medium used was α-minimum essential medium (α-MEM) (12000–022; Gibco) supplemented with AlbuMAX I (11020–021; Thermo Fisher Scientific) at a final concentration of 40 mg/mL. $NaHCO_3$ (7%) was then added (0.026 ml/ml medium) to achieve a final concentration of 1.82 g/L (0.0182 g for 10 ml of medium). Antibiotic-antimycotic (15240062; Thermo Fisher Scientific) was added at a 1/100 volume to achieve a final concentration of 100 IU/mL for penicillin, 100 μg/mL for streptomycin, and 250 ng/mL for amphotericin. Sterilization was performed with Millipore filtration and followed by storage in a refrigerator before use.

## Agarose gel preparation

To make the agarose gel block for organ culture, agarose powder (Dojindo Molecular Technologies) was dissolved in water purified with Milli-Q (35–1251; Merck) at 1.5% (w/v) and autoclaved. While cooling, 33 mL of the agarose solution was poured into 10 cm dishes to form a 5 mm thick gel. The gel was cut into squares of approximately $10 \times 10$ mm size and these were used as a stand for testis tissue placement. The gel blocks were submerged in the culture medium in 12-well culture plates for more than 6 h before use. Testes tissue fragments or isolated STs were transferred to the surface of agarose gel blocks, the lower half of which were soaked in 0.5 mL medium in individual wells. A medium change was performed once a week. The culture incubator was supplied with 5% carbon dioxide in air (20% $O_2$), unless otherwise described in the text, and maintained at 34˚C.

## PDMS ceiling chip

The PDMS (polydimethyl-siloxane) ceiling (PC) chip was fabricated by conventional photolithography and soft lithography methods [15]. The PC chip was produced by mixing PDMS prepolymer and curing reagent (Silpot 184; Dow Corning) at a 10:1 weight ratio. The mixture was poured into a mold master made by the photolithography method, and the mold master was placed in a vacuum chamber for degassing. After 30–45 min, the mold master was moved to an oven and baked for 2 h at 70˚C. After cooling down, the solidified PDMS was peeled off from the master and this PDMS disk was cut into individual chips with a cutting knife. The thickness of the chips was dependent on the amount of PDMS mixture poured in. The depth of dent, the tissue setting area, ranged from 150 μm to 190 μm depending on the mold [16]. The modified PC chip, named the STPC chip, had a row of pillars in the tissue set region. The pillars were 0.1 mm wide squares, equal in height to the depth of the set region (150–190 μm), and arranged in straight rows at 0.1 mm intervals (Fig 1D). A thin PDMS membrane (75 μm thick, ASAHI Rubber Inc.) was used in some cases for pressing testis tissues against the base agarose gel by placing them under the PDMS ceiling to hold the tissue in place.

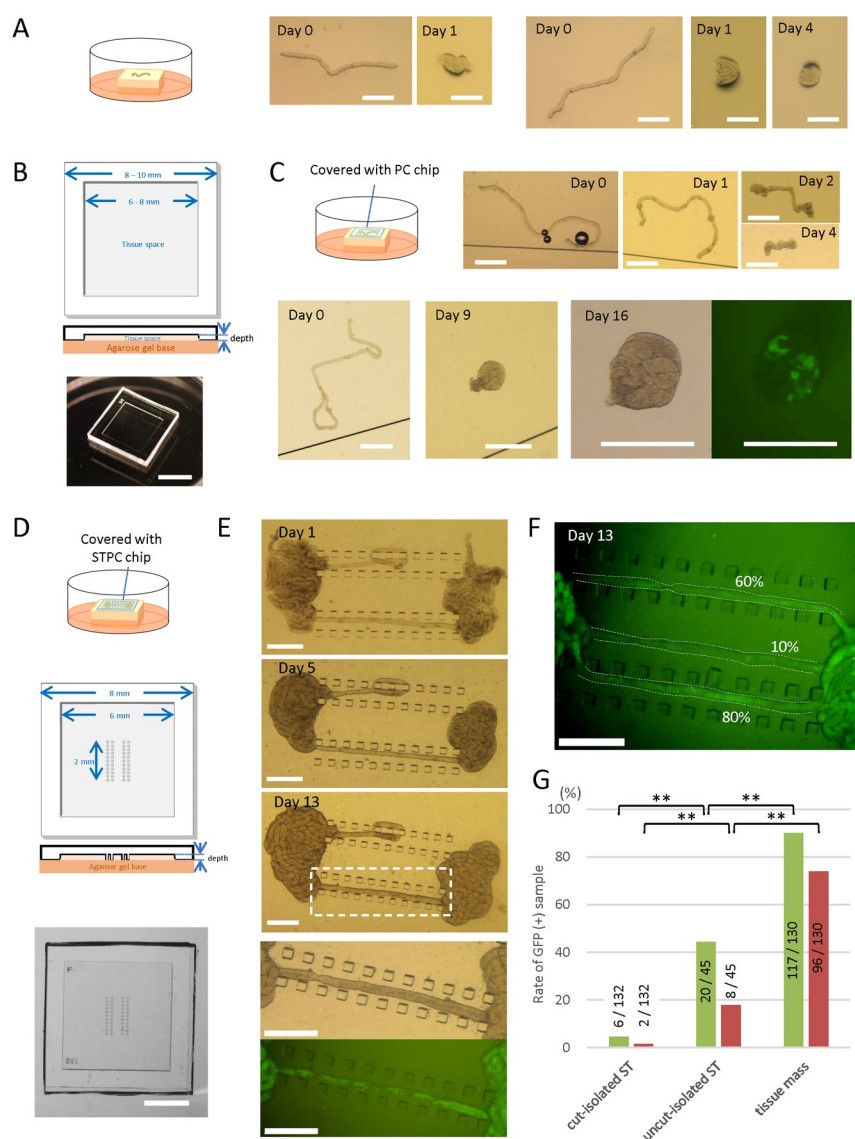

**Fig 1. Culturing of isolated ST segments.** (A) According to an air-liquid interphase method, cut-isolated STs of a 5 dpp mouse were placed on an agarose gel block half-soaked in the medium. The STs shrunk and conglomerated in a day. (B) Schematic diagram of a PC chip, top and cross-sectional views, and photos of a PC chip. The depth of dent for accommodating culture samples was between 150 and 190 μm. (C) A schematic drawing of a culture experiment using a PC chip, and a stereomicroscopic view of a cut-isolated ST of a 5 dpp mouse covered with a PC chip. The ST shrunk in several days. Another cut-isolated ST cultured under a PC chip conglomerated but showed GFP expression on culture day 16. (D) A schematic drawing of a culture experiment using STPC, and a schematic diagram of a STPC chip, top and cross-sectional views. The size of each pillar was 0.1 × 0.1 mm. Four parallel pillar rows made of 11 pillars each were arranged in the center of the tissue space. The depth of the tissue space ranged from 150 to 190 μm. A photo of an STPC chip from above is shown in the bottom pannel. (E) Stereomicroscopic view of an uncut-isolated ST cultured under an STPC chip on days 1, 5, and 13. The dashed rectangular area is enlarged in the bottom two panels. Fluorescence microscopy revealed *Acr*-GFP expression in the stretched portion of the ST. (F) A photo of the GFP-expressing portion of three uncut-isolated STs, taken on culture day 13. The percentages of GFP-expressing area were visually measured as 60%, 10%, and 80%, respectively. (G) The percentages of GFP-positive area in the samples of three ST states: cut-isolated STs, uncut-isolated STs, and tissue mass. Samples with GFP-expressing areas over 10% (green bar) and 50% (red bar) during culture days 31 to 40 were counted as GFP-positive. Numerators and denominators in or over each bar are the numbers of GFP-positive tissue samples and total examined tissue samples, respectively. The experiments were performed under a 20% $O_2$ concentration. **$P<0.01$. Scale bars: 5 mm (B), 2 mm (D), 0.5 mm (A, C, E).

## Culture methods–PDMS ceiling (PC) method

The testes of mice were decapsulated—i.e., the capsule (tunica albuginea) was removed by forceps. The remaining tissue mass was mostly composed of seminiferous tubules, which lightly adhered to each other via the intervening microvasculature. Each testis tissue was gently separated into 6–8 pieces with forceps. For tissue mass culture, these clamps were simply placed on the agarose gel block. For the culture of isolated STs, the testis tissue mass was soaked in the medium to loosen and disperse the STs. The STs were then gently separated with forceps and cut with spring scissors as needed, depending on the desired state of the STs. These STs in the medium were sucked up with a micropipette and poured in the hollow space of a PC chip (Fig 1B). The PC chip was then inverted and placed on an agarose gel. In case of the uncut-isolated ST method—i.e., to separate the STs without cutting—the tissue mass was placed in the hollow portion of the STPC chip (Fig 1D). Forceps were used to gently dissociate the tissue mass into STs, and then to isolate an ST segment with both ends connected to the tissue mass so that the ST segment would fit between the pillar rows. The STPC chip was then inverted and set on an agarose gel.

## Culture methods—Soft agarose method

After preparing cut-isolated STs in the culture medium, an equivalent volume of low melting point Agarose-L (319–01181; Nippon Gene) solution (1% w/v) was added and gently mixed. After cooling down at 4°C for about 15 min, the soft agarose gel containing the cut-isolated STs was cut into squares of 8 × 8 mm size and laid on a 1.5% agarose gel block (base gel). Medium was poured to half the height of the gel block (S3 Fig, Method A in S1 File and S1 Video). The PDMS ceiling chip was optional but was used in many cases. As an alternative approach, we adopted the protocol of a soft agar culture system from a previous study [8]. In brief, the soft agarose (0.5% w/v) containing the cut-isolated STs was poured onto the solid bottom gel layer. To make the bottom gel layer, 1.5% agarose solution was poured into 5 cm or 3 cm dishes. After cooling down and solidification, in order to replace the water in the gel with medium, an identical volume of culture medium was added to the dish and left for more than 6 h until use. The dish with its 2 layers of gel was left at 4°C for about 15 min until the soft agar layer became solid. The upper soft gel layer was adjusted to a thickness of around 0.4 mm (exactly 0.395, 0.443, 0.435, 0.220, and 0.361 mm when measured). Then the two-layer agarose gel was cut into squares of about 10 × 10 mm size, each of which was moved to a well in a 12-well plate with 0.5 mL medium in each well (S3 Fig, Method B in S1 File and S1 Video). Medium change was performed once a week. The culture incubator was supplied with 5% carbon dioxide in air and maintained at 34°C.

## Observations

Cultured tissues and ST segments were observed at least once a week under a stereomicroscope equipped with an excitation light for GFP (LeicaM205 FA; Leica, Germany). *Acr*-GFP begins to be expressed in mice at around 15 dpp *in vivo*, but its expression could be delayed by several days *in vitro*. In addition, GFP emission can fluctuate at intervals of several days. Therefore, to avoid false-negative results, GFP positivity was not necessarily determined on a single day; in most cases, two observations with a 7-day interval were taken, resulting in an observation period of greater than 7 days. Samples showing GFP expression during those periods were considered GFP-positive. The GFP-positive portions in each stretch of ST were measured by visual approximation as 0%, 1–10%, 11–20%, 21–40%, 41–60%, 61–80% or 81–100%.

H3.3 mCherry appears beginning at around 28 dpp *in vivo*. *In vitro*, however, its expression is delayed days to weeks, depending on the sample. For the reliable identification of mCherry,

each sample tissue was removed from the culture well, placed on a slide glass and observed with an inverted microscope (IX73; Olympus) or a confocal microscope (FV1000–MPE; Olympus). The observation timing was carefully determined in each case, taking the status of preceding GFP expression into account.

## Histological and immunohistochemical examinations

In histological examinations, specimens were fixed with Bouin's fixative and embedded in paraffin. One section showing the largest cut surface was made for each specimen and stained with hematoxylin and eosin (H&E) or periodic acid Schiff (PAS). In immunofluorescence staining, tissues were fixed with 4% of paraformaldehyde in PBS at 4˚C overnight. Tissues were then soaked in solutions of 10%, 15%, and 20% (w/v) sucrose in PBS for 1 h each in succession for cryoprotection. They were cryo-embedded in OCT compound (Sakura Finetek Japan) and cut into 7-μm-thick sections. Antibodies used as primary antibody were anti-GFP (ab13970, 1:1000; abcam), anti-synaptonemal complex protein 3 (SCP3) (ab97672, 1:100; abcam), and anti-GFRα1 (1:200, AF560; Bio-Techne, MN, USA). Lectin PNA from Arachis hypogea (peanut), Alexa Fluor 568 Conjugate (L32458, 1:1000; Thermo Fischer Scientific, MA, USA) was used to identify the acrosome. Antibodies used for secondary antibody were Alexa Fluor 488-conjugated goat anti-chicken antibody (A-11039, 1:200; Thermo Fischer Scientific), Alexa Fluor 555-conjugated goat anti-mouse antibody (A-21424, 1:200; Thermo Fischer Scientific), Alexa Fluor 555-conjugated goat anti-rabbit antibody (A-21428, 1:200; Thermo Fischer Scientific), Alexa Fluor 647-conjugated goat anti-rat antibody (A-21247, 1:200; Thermo Fischer Scientific), and Alexa Fluor 647-conjugated donkey anti-goat antibody (A-21447, 1:200; Thermo Fischer Scientific). Nuclei were counterstained with Hoechst33342 dye. Observation of immunostained samples were performed with a confocal laser microscope (FV1000-MPE; Olympus).

## Whole-mount immunohistochemical staining

The cultured seminiferous tubules were fixed with 4% paraformaldehyde in PBS at 4˚C overnight. After washing with PBS for 3 min, the tubules were dehydrated in 100% methanol for 30 min at room temperature. The dehydrated tubules were washed with PBS containing 1% Triton-X100 (1% PBST) 4 times for 10 min each, and then blocked with Image-iT™ FX Signal Enhancer (Thermo Fisher Scientific) for 1 h and incubated with primary antibody diluted in blocking buffer at 4˚C overnight. After washing with 1% PBST 4 times for 10 min each, the tubules were reacted with secondary antibody diluted in blocking buffer at room temperature for 1 h. After washing with 1% PBST 4 times for 10 min each, the nuclei were counterstained with Hoechst 33342 dye. Specimens were observed with a confocal laser microscope (Olympus FV-1000D). The primary antibodies used were chicken anti-GFP antibody (1:1000; Abcam), rabbit anti-RFP polyclonal antibody (1:1000; MBL), mouse anti-SYCP3 antibody (1:500; Abcam), goat anti-GFRα1 antibody (1:200; R&D Systems), rabbit anti-STRA8 antibody (1:250; Abcam), rabbit anti-γH2AX (1:500; Abcam), and rabbit anti-Mouse HSD3B antibody (1:250; Trans Genic Inc.). The secondary antibodies used were goat anti-chicken IgG, goat anti-rabbit IgG, and goat anti-mouse IgG, conjugated with Alexa 488, Alexa 555 or Alexa 647 (1:200; Invitrogen).

## Statistical analysis

Fisher's exact test was used for statistical analyses. The Holm method was used to adjust the family-wise error rate in multiple comparisons.

## Results

### *Acr*/H3 double-Tg mouse testis as a system for monitoring spermatogenesis

The GFP expression in the spermatogenic cells of *Acr*-GFP transgenic mice starts from pachytene spermatocytes at stage 4 onward [12]. In Histone H3.3-mCherry transgenic mice, mCherry is expressed in spermatids from step 11 onward [14]. Then, we first examined if *Acr*/H3 double-Tg mouse testis faithfully express GFP and mCherry as expected in their progression of spermatogenesis. Testes of mice aged 10, 20, and 40 days postpartum (dpp) were collected and prepared for immunohistochemical observation (S1A Fig in S1 File). The testis of 10 dpp was negative for both GFP and mCherry. In the 20 dpp testis, GFP-positive cells and peanut agglutinin (PNA)-positive dots were observed in the center of STs, indicating haploid cells were emerging. In the 40 dpp testis, GFP, mCherry and PNA were clearly observed. At higher magnification, the combination of GFP and mCherry, which respectively localized to the acrosome and nucleus, was identified as a unique figure that served as a faithful discriminative marker of elongating spermatids at step 11 onward (S1B Fig in S1 File) [14]. These results confirmed that the Acr/H3 double-Tg mouse testis was an excellent system for monitoring the progression of spermatogenesis and spermiogenesis.

### Cut and isolated ST segments shrunk in cultivation

After removing the tunica albuginea, STs in the testis of a neonatal *Acr*-GFP Tg or an *Acr*/H3 double-Tg mouse were loosened and dispersed, in order to randomly cut and isolate ST segments (cut-isolated STs). The cut-isolated STs were cultured on an agarose gel block, which was half-soaked in culture medium, and the GFP expression was monitored with a stereomicroscope. However, these isolated stretches of ST shrunk and conglomerated in a day (Fig 1A). We suspected that this may have been due to the surface tension of the medium, and we therefore used a ceiling chip made of an air-permeable silicone, polydimethylsiloxane (PDMS), to cover the ST segments [16, 17]. This PDMS ceiling (PC) chip is a small PDMS plate having a shallow dent/hollow on one side which serves as a space for testis tissues and STs (Fig 1B). By placing the PC chip over the ST, we expected it could alleviate the surface tension and reduce the shrinkage. This strategy worked to some extent, but within several days the cut-isolated STs beneath the chip had gradually contracted and shrunk (Fig 1C). Proper spermatogenesis could not be expected in such shrunken STs. In fact, among the 132 cut-isolated ST samples that exhibited shrinkage in culture, GFP-expression was observed in only 7 (5.3%) (Fig 1C and 1G). We then produced a new PC chip, named the seminiferous tubule PC (STPC) chip. In the dent space of the STPC chip, we designed four rows of aligned pillars to hold the ST in place, hoping to prevent or reduce ST shrinkage (Fig 1D and S2A Fig in S1 File). However, we found that the cut-isolated STs set between the pillar rows contracted to almost the same degree as those without pillar rows (S2B Fig in S1 File). Even among several STs that we set in ways to hold them, such as meandering between the pillars, shrinkage still occurred (S2C, S2D Fig in S1 File). Such ST shrinkage would be expected to disturb and obstruct the progression of spermatogenesis.

### Uncut and segmentally isolated ST method

In order to prevent the shrinkage of STs, we gave up cutting and detaching the STs completely, and instead isolated a portion of the ST, 1–2 mm in length, leaving both ends connected to the tissue mass. The ST segment isolated in this manner, which we designated the uncut and segmentally isolated ST (uncut-isolated ST), was placed in the STPC, and the tissue masses were placed on both sides of the edge of the pillar rows (Fig 1E). Although the tissue as a whole

again showed a tendency to contract, the segmentally isolated part of the ST remained stretched. After 2 weeks of culture, GFP expression was observed in the uncut-isolated STs by fluorescence microscopy (Fig 1E). Photographs were taken weekly and the ratio of the GFP-expressing portion to the total uncut-isolated ST was measured visually (Fig 1F). The uncut-isolated STs whose GFP-expressing portion constituted more than 10% or 50% of the total length were regard as GFP-positive. We then compared STs in 3 different states: cut-isolated STs, uncut-isolated STs and tissue masses (Fig 1G). For cut-isolated (conglomerated) STs, only 5.3% of samples were GFP-positive with a 10% positive area threshold, as stated above. With a 50%-positive area threshold, only 1.5% (2 out of 132) of samples were positive. The uncut-isolated STs showed GFP-positivity rates of 53.3% (24 out of 45 samples) and 31.1% (14 out of 45 samples) for the 10% and 50% positive-area thresholds, respectively. The tissue masses showed 90.0% (117 out of 130 samples) and 73.8% (96 out of 130 samples) positivity rates, respectively. These results indicated that although spermatogenesis was induced in cut-isolated and uncut-isolated STs, at least up to the *Acr*-GFP-expressing stage—i.e., the early pachytene stage [13]—the efficiency of spermatogenesis was lower than that in the tissue mass.

## Long-term culture of uncut-isolated STs

When we performed a culture experiment of uncut-isolated STs for a long period with frequent observations, *Acr*-GFP expression in a ST showed unique oscillatory fluctuation (Fig 2A and 2B). The same phenomenon was observed and reported previously in our study with a tissue mass culture experiment using a microfluidic device [18]. We conjectured that this fluctuation reflected an incomplete spermatogenesis in which degenerated and dead *Acr*-GFP expressing cells were cleaned up by phagocytotic Sertoli cells, followed by the reappearance of *Acr*-GFP expressing cells from among the remaining precursor cells in the ST. This finding thus indicated that the isolated ST can support spermatogenesis, although incompletely, including not only the so-called first wave but also subsequent successive spermatogenesis. To determine how far spermatogenesis had proceeded in the uncut-isolated STs, we performed whole-mount immunochemistry on STs from a 5 dpp mouse cultured for 34 days, with antibodies to SYCP3, GFP, and mCherry. Cells positive for SYCP3 and GFP, representing meiotic spermatocytes at the pachytene stage, were observed as the most advanced differentiating cells (Fig 2C).

## Cultivation of cut-isolated STs using a soft agarose method

As a second approach to prevent the shrinkage of STs, we adopted a soft agarose method. Initially, 0.35% soft agarose was used as described in the literature [8, 19] and the cut-isolated STs mixed in the gel were set in the STPC chip and placed on the base agarose gel (1.5%). Under this condition, however, the shrinkage was not sufficiently suppressed. We therefore raised the concentration of the soft agarose in the gel to 0.5%. A square thin piece of soft agarose containing the cut-isolated STs was moved onto the base gel, with or without a PC chip covering (S3 Fig, Method A in S1 File and S1 Video). However, the soft agarose gel was too fragile to handle by itself, and the gel was prone to breakage during the transfer. We therefore adopted Gholami's method [8]. The cut-isolated STs suspended in 0.5% soft agarose solution were poured on the bottom layer of a 1.5% agarose gel, and the resultant two-layered gel was cut out for cultivation (S3 Fig, Method B in S1 File and S1 Video). The cut-isolated STs in 0.5% soft agarose gel did not shrink and maintained their original form (Fig 3A). In this procedure, in addition to cut-isolated STs, small clumps of aggregated STs were observed in the gel. These mostly resulted from the insufficient isolation of STs and were denoted as aggregated STs when 3 or more STs clustered together (Fig 3B).

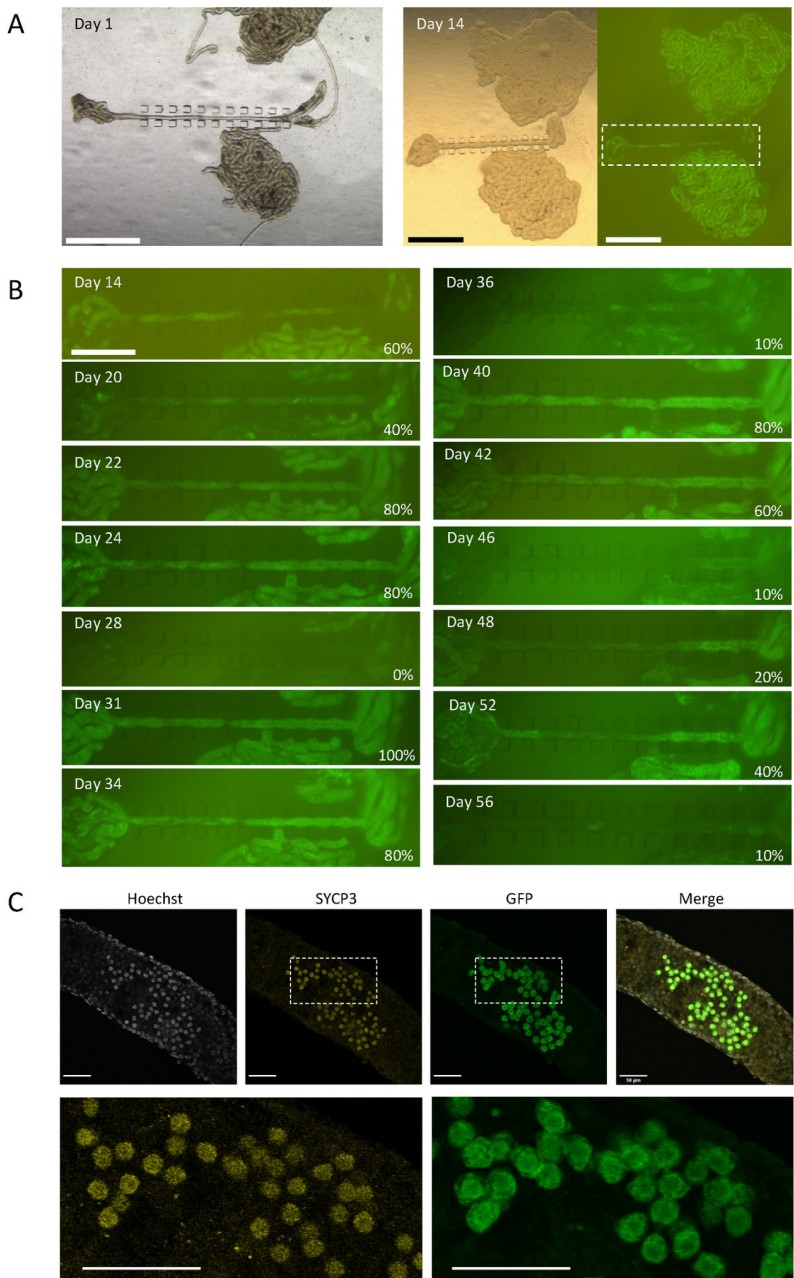

**Fig 2. A long-term culturing of uncut-isolated STs using STPC chips.** (A) A testis of an *Acr*-GFP mouse of 3 dpp were dissected and STs were set under STPC chips for culturing. On culture day 14, GFP expression was observed. The dashed rectangular region is enlarged in B. (B) Photographs taken sequentially on each culture day under excitation light for GFP recognition were aligned. GFP expression rates are noted in the lower right corner of each panel. (C) Confocal microscopic photos of whole-mount immunochemical staining of an uncut-isolated ST were taken on culture day 34, using a 5 dpp mouse testis. SYCP3- and GFP-positive cells represent meiotic spermatocytes. Scale bar: 1 mm (A), 0.5 mm (B), 50 μm (C).

Within two weeks, GFP expression was confirmed in 72 out of 268 cut-isolated STs (26.9%) under fluorescence microscopic observation (Fig 3A, 3C and 3E). As for aggregated STs, GFP expression was observed in 55 out of 90 samples (61.1%) (Fig 3D and 3E). This result suggested that STs under the aggregated condition were somehow favored for spermatogenic

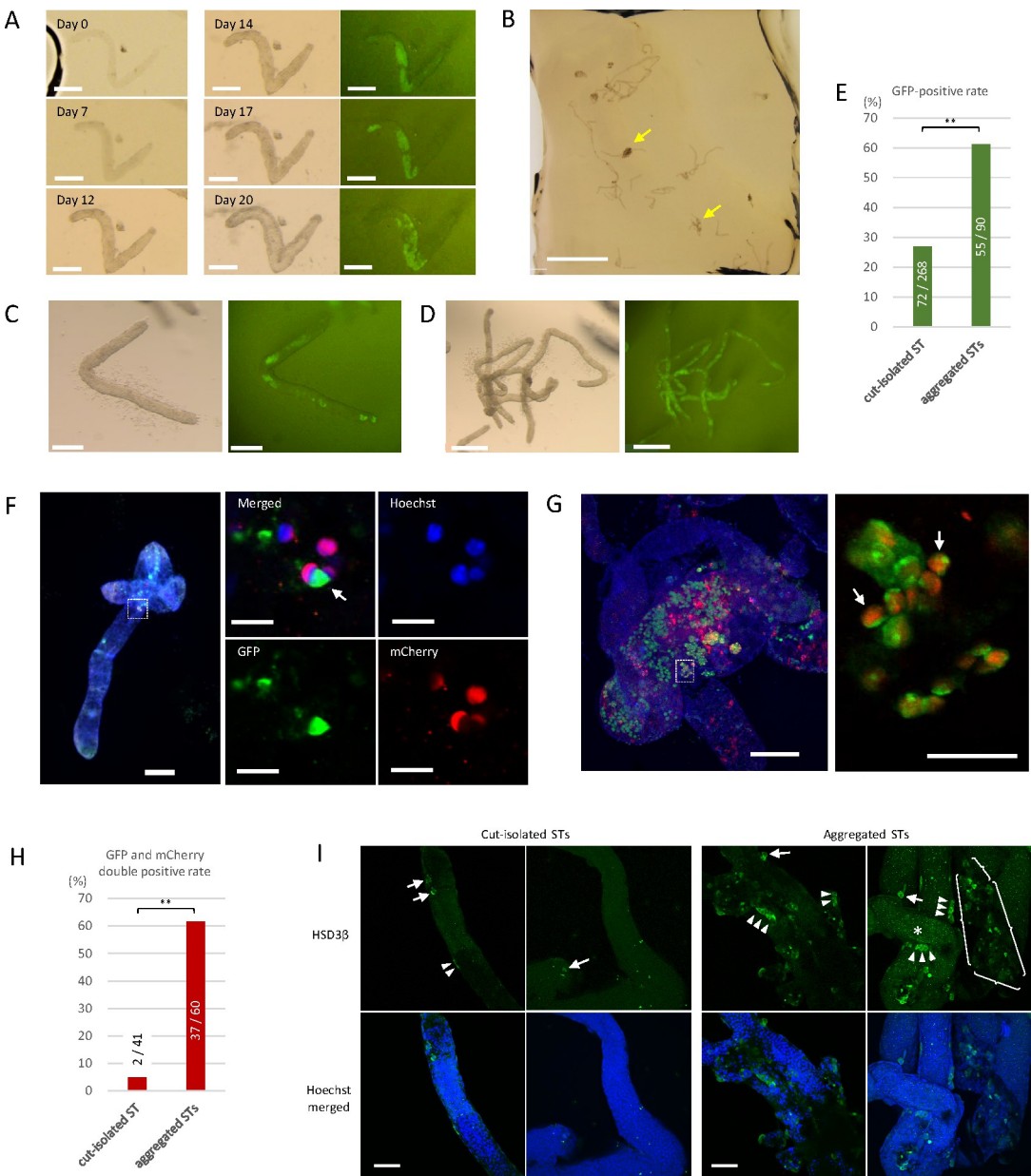

**Fig 3. Culture of cut-isolated and aggregated STs in the soft agarose.** (A) Series photos of a cut-isolated ST embedded in soft agarose using method B, showing the maintenance of its form without shrinkage and the induction of GFP expression from day 14. (B) Low-magnification view of cut-isolated STs embedded in the soft agarose. Yellow arrows indicate aggregated STs. (C) A cut-isolated ST in the soft agarose with GFP expression at culture day 22, derived from a 7 dpp mouse testis. (D) Aggregated STs in the soft agarose with GFP expression at culture day 22, derived from a 7 dpp mouse testis. (E) GFP expression was examined by fluorescence microscopy in samples from 7 experiments and assessed during culture days 20 to 25. Numerators and denominators in or over each bar are the numbers of GFP-positive tissue samples and total examined tissue samples, respectively. The experiments were performed under a 20% $O_2$ concentration. ** P < 0.01. (F) Whole-mount immunostaining of a cut-isolated ST with anti-GFP antibody (green) and anti-RFP antibody (red). Nuclei were counterstained with Hoechst 33342 dye (blue). The box indicated by the dotted lines in the left panel is enlarged in the right panels. Germ cells with both GFP and mCherry expressions, indicating elongating spermatids at or beyond the step 11, were observed (white arrow). (G) Whole-mount immunostaining of aggregated STs demonstrated several elongating spermatids, expressing both GFP and mCherry. The rectangular region surrounded by a dotted line is enlarged in the right panel. (H) GFP and mCherry expressions were confirmed by whole-mount immunohistochemical staining in samples from a single experiment. The tissues were fixed either on culture day 45, 52, or 59. The experiment was performed under a 20% $O_2$ concentration. ** P < 0.01. (I) Whole-mount immunostaining of cut-isolated and aggregated STs with HSD3β antibody (Green), along with counterstaining with Hoechst. Within aggregated STs, faint dot-like GFP expression derived from Acr-GFP was observed (*). An HSD3β signal was observed

in cells adhering to the outside of the STs, either singly (arrows) or as clusters (arrowheads). In these pictures, the numbers of such HSD3β-positive cells were counted to be 5 in cut-isolated and 41 in aggregated STs. In addition, in the case of aggregated STs, a clumpy interstitial tissue containing about 20 HSD3β-positive cells was also observed (surrounded by braces). Scale bars: 200 μm (A, C, F left, G left), 2 mm (B), 0.5 mm (D), 50 μm (G right, I) and 10 μm (F right 4 panels).

progression. In 40–55 days, STs from an experiment were examined by whole-mount immunochemical staining. In 2 of the 41 samples of cut-isolated STs examined, cells positive for both GFP and mCherry were observed (Fig 3F and 3H). This result showed that cut-isolated STs can support spermatogenesis up to the formation of elongating spermatids. As for aggregated STs, GFP and mCherry double-positive cells were observed in 37 out of 60 samples examined (Fig 3G and 3H). Again, it appeared that aggregation of STs confers an advantage for the promotion of spermatogenesis.

We speculated that interstitial cells that remain attached to the aggregated STs could be a reason for the abundant and more advanced spermatogenesis. To investigate this possibility, we stained Leydig cells with anti-HSD3β antibody under a whole-mount condition. In this instance, HSD3β-positive cells were detected abundantly in aggregated STs and only marginally in isolated ST samples (Fig 3I). The relationship between spermatogenic efficiency and the amount of interstitial cells around the ST would be an important issue to scrutinize in future studies.

## Lowered oxygen concentration was favorable for spermatogenesis

In our previous study using rat testis tissues, we found that an $O_2$ concentration lower than the atmospheric concentration of 20% induced spermatogenesis with higher efficiency [20]. Then, in a pilot study using testis tissues of *Acr*-GFP Tg mice, we compared two different concentrations of $O_2$, i.e., 20% and 10%, in the incubator (S4 Fig in S1 File). In 20% $O_2$, GFP expression was observed evenly throughout the tissue. On the other hand, in 10% $O_2$, GFP expression was limited exclusively to the peripheral regions, leaving the central region completely negative. This expression pattern was reasonable, considering the limited diffusion of $O_2$ into the tissue mass in the case of 10% $O_2$. However, round spermatid formation, identified by GFP aggregation into a cap-shape, was intensively observed in the GFP-expressing area of 10% $O_2$ samples. Round spermatids were also observed in 20% $O_2$ samples, but rather sporadically and as smaller foci. Because the physiological $O_2$ concentration in a body is around 1% to 8% [21], a concentration of 20% $O_2$ must be too high for many, if not all, cellular activities.

The above results suggested that there was a benefit in culturing the testis tissue in 10% rather than 20% $O_2$. However, the regional difference in spermatogenic progression in a single tissue mass made it difficult to interpret the effect of oxygen concentration. After all, the oxygen tension can vary significantly from one location to another within a tissue mass, and the tension can continuously change in accordance with the oxygen permeation throughout the tissue and consumption by the tissue.

We therefore adopted the uncut-isolated ST culture method to examine the effect of $O_2$ concentrations of 20%, 15% and 10% on the spermatogenesis (Fig 4A). Among the 36 samples cultured in 20% $O_2$, the number having an *Acr*-GFP-expressing region rate over 10% and 50% in a stretch of ST were 16 (44.4%) and 3 (8.3%), respectively. The corresponding numbers among the 41 samples cultured in 15% $O_2$ were 31 (75.6%) and 10 (24.4%), respectively. Among the 27 samples cultured in 10% $O_2$, the respective values were 25 (92.6%) and 21 (77.8%) (Fig 4B). The number of samples varied among the three groups due to technical difficulties in the preparation of uncut-isolated STs. Namely, some samples were lost prior to evaluation due to fragmentation or slipping of the ST apart from the frame of the PDMS pillars.

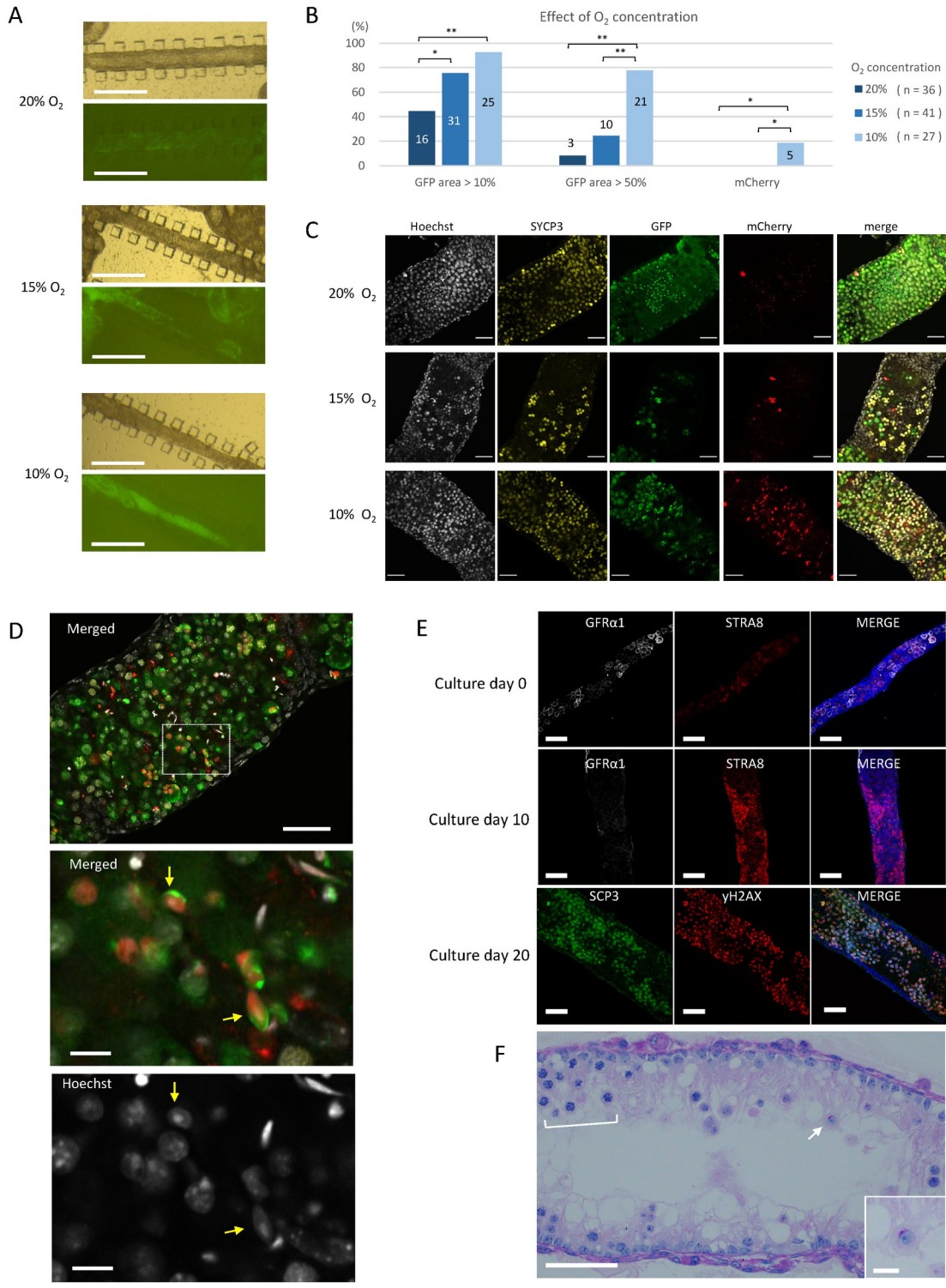

**Fig 4. Effect of lower oxygen concentrations on spermatogenesis evaluated with uncut-isolated STs.** (A) Stereomicroscopic view of uncut-isolated STs cultured at different $O_2$ concentrations. GFP expression was observed at culture day 25 by a fluorescence microscope. (B) Data from nine experiments are summarized. Samples with GFP-expressing areas over 10% and 50%, respectively, during culture days 20 to 30 were counted as GFP-positive. The mCherry expression was confirmed on culture day 30 or 36. Numbers in or over the bar are the numbers of marker-positive samples among total samples examined in each $O_2$ concentration; the numbers of total sample are shown in parenthesis. * P < 0.05, ** P < 0.01. (C) Whole-mount immunostaining with anti-SYCP3 antibody (yellow), anti-GFP antibody (green), and anti-RFP (mCherry) antibody (red) was performed on culture

day 34, using a 5dpp Acr/H3 double-Tg mouse testis. (D) A 5dpp Acr/H3 double-Tg mouse testis was cultured in 10% $O_2$ for 34 days and examined by whole-mount immunohistochemistry. A merged image of GFP (green), mCherry (red), and Hoechst (gray) was shown in the top panel. The dashed rectangular area is enlarged in the panels below. An acrosome cap (green) and mCherry-positive nuclei (red) of elongated spermatids (yellow arrows) were observed. (E) Uncut-isolated STs cultured under 10% $O_2$ for 0, 10 and 20 days were subjected to whole-mount immunostaining. (F) An uncut-isolated ST cultured under 10% $O_2$ for 20 days was sectioned and stained with PAS, revealing the regional layers of spermatocytes (bracket) and an isolated round spermatid (arrow and enlarged in the inset) having a cap-shaped acrosome stained in red. Scale bars: 500 μm (A), 50 μm (C, D top, E, F), and 10 μm (D middle and bottom, F inset).

Nonetheless, this experiment clearly demonstrated that lower $O_2$ concentrations were associated with a higher rate of GFP expression in a larger area. Interestingly, such a difference was not obvious among tissue masses cultured in the same respective STPC chips. The tissue masses showed *Acr*-GFP expression in most of their regions regardless of $O_2$ concentration (S5 Fig in S1 File). This relative insensitivity of the tissue masses to $O_2$ concentration could have been caused by reduced $O_2$ tension in a large area of tissue mass due to the consumption of $O_2$ by the tissue itself.

After culturing about 30 days, 6 GFP-positive samples, 2 in each $O_2$ concentration group, were examined with whole-mount immunochemical staining. Sycp3-positive cells (meiotic cells) and GFP-positive cells were confirmed in all the $O_2$ concentration groups. However, mCherry-positive cells were sparse in samples cultured in 20% and 15% $O_2$, but the samples cultured in 10% $O_2$ contained many such cells (Fig 4C). Germ cells expressing both GFP and mCherry, representing elongating spermatids at step 11 onward, were confirmed only in the samples in 10% $O_2$, and only in 5 of 27 samples (Fig 4D and 4B).

To ensure that this condition—namely uncut-isolated ST under 10% $O_2$—recapitulates the normal development of the first wave spermatogenesis, a time-course examination using whole-mount immunohistochemistry with stage-specific markers was performed (Fig 4E). The STs of a 5 dpp mouse contained undifferentiated spermatogonia, which were stained positive with GFRα1, as the most abundant germ cell at explantation (day 0). In 10 days, STRA8-positive cells, probably preleptotene spermatocytes, emerged and increased in number, while GFRα1-positive undifferentiated spermatogonia became more sparse. In 20 days, spermatocytes positive for both SCP3 and γH2AX appeared and the STs became markedly thicker. A sectioned histological examination with PAS staining showed spermatocytes and round spermatids (Fig 4F). Taken together, these results demonstrated that the 10% $O_2$ concentration was more favorable than higher $O_2$ concentrations for promoting spermatogenesis up to the production of elongating spermatids. Of note, it is even possible that $O_2$ lower than 10% would be optimal for isolated STs to promote spermatogenesis. In conclusion, culturing isolated STs is a sensitive and reliable means for elucidating the microenvironmental signals that affect or control spermatogenesis.

## Discussion

In the history of mammalian *in vitro* spermatogenesis studies, culturing isolated segments of ST can be categorized as a separate technique from culturing tissue as a mass [5]. The first studies using the former culture technique were performed in the 1970s [22]. However, Parvinen and colleagues would be the first to report the progression of spermatogenesis in a defined segment of adult rat STs [6]. Specifically, STs containing late pachytene and diakinetic primary spermatocytes were cultured in a chemically defined medium for 6 days. In those STs, meiotic divisions were completed and the newly formed spermatids, which had acrosonic systems characteristic of step 5, were observed [6, 7]. In a subsequent study, Toppari *et al*. extended these findings. They excised segments of rat STs from stages II to III and cultured them in the

same way. The round spermatids at steps 2–3 developed into step 7 spermatids within 7 days, replicating the progression *in vivo*. Moreover, the spermatogonia and spermatocytes in that segment developed in correspondence with those *in vivo* [23].

We initiated the present study as an extension of our previous studies of testis tissue cultivation, which succeeded in the production of fertile sperm from spermatogonial stem cells [4, 24]. To our surprise, the STs underwent drastic shrinking and lost their original tubular appearance in only a couple of days. However, the explanted tissues themselves would certainly have experienced drastic changes in environmental conditions, and this would be expected to elicit various reactions in the tissues. In fact, we recently observed that testis tissues exhibited a significant level of inflammation within a mere two days after explantation [25]. After trying several approaches, we resorted to isolating STs segmentally without cutting them in order to keep them from shrinkage. This was an incomplete method of isolation, but it was an effective method for inducing spermatogenesis at a higher efficiency. Having settled on this uncut-isolation method, we then explicitly evaluated the effect of oxygen concentration on spermatogenesis. In our pilot study we cultured tissue masses under different $O_2$ concentrations, and the results suggested that lower oxygen concentration might be superior in inducing efficient spermatogenesis. However, because the oxygen tension at any point in a tissue is the result of the sequential oxygen supply and consumption at that point as well as at points nearby, the value fluctuates and is difficult to measure precisely. The results of the pilot study were thus hard to interpret in a straightforward manner. In contrast, the segmentally isolated portion of STs should be bathed in a constant $O_2$ concentration that is close to the concentration set in the incubator throughout the cultivation period. Therefore, culturing STs, instead of tissue masses, seemed to be a superior way to directly evaluate the effect of environmental factors.

In addition to performing uncut-isolated ST experiments, we also successfully minimized the shrinkage of cut-isolated STs by adopting the soft agarose method reported by Gholami and colleagues [8]. The soft agar culture method was initially developed to characterize clonal expansion of bone marrow cells [26–28]. Then, it was widely applied in different ways, and eventually became the gold-standard assay for cellular transformation *in vitro* [29–32]. In 2008, Stukenborg *et al.* used the soft agar method to cultivate spermatogonia for proliferation. They also intended to induce differentiation of spermatogonia by coculturing with somatic cells [19]. Gholami's report was the first trial, to our best knowledge, to culture STs in soft agar [8]. Although they reported the progression of spermatogenesis, their evaluations were incomplete, and thus further studies were warranted. In the present study, we used *Acr*/H3 double-Tg mice which faithfully express GFP and mCherry, respectively, as reliable markers of meiotic pachytene and step 11 spermatids onward. In particular, the combination of GFP accumulation in the acrosome and mCherry expression in the nucleus, which were located closely side by side, was a strong marker for the identification of elongating spermatids. Thus, it was demonstrated faithfully for the first time that *in vitro* spermatogenesis in the isolated STs proceeded up to the stage of elongating spermatids.

Although our experiments demonstrated that culturing ST segments without shrinkage is possible, the efficiency was much lower than that by culturing tissue masses or aggregated STs. The GFP-positive rates for each isolation method were summarized as 5.3%, 26.9%, 53.3%, 61.1%, and 90% for cut-isolated conglomerated, cut-isolated in soft agarose, uncut-isolated, aggregated in soft agarose, and tissue mass, respectively, under 20% $O_2$ concentration (S6 Fig in S1 File). This indicates that STs that remained adherent rather than isolated singly showed better performance in inducing spermatogenesis. On the other hand, when the uncut-isolated method was performed under 20%, 15%, and 10% $O_2$, the GFP positive rates were 44.4%, 75.6%, and 92.6%, respectively. This result suggests that the local $O_2$ tension, which should be reduced in aggregated STs and tissue masses, contributes significantly to spermatogenesis

performance. However, there are two other possible explanations for the favorable performance of aggregated STs and tissue masses. First, interstitial cells attaching to the STs could be essential or at least influential. Aggregated STs and tissue masses would be expected to contain interstitial cells more abundantly than isolated STs. Indeed, our whole mount immunohistochemical study showed that the aggregated STs contained many Leydig cells, while the cut-isolated STs contained only a few. In addition to Leydig cells, there are several different types of cells outside the STs in the testis, including macrophages, lymphatic endothelial cells and so forth. Numerous studies have described the pivotal roles of these cells in spermatogenesis. Leydig cells produce and secrete androgens and other factors, such as colony-stimulating factor 1 (CSF-1) [33]. The testicular macrophages also produce CSF-1 and enzymes involved in retinoic acid (RA) biosynthesis [34]. Lymphatic endothelial cells secrete fibroblast growth factors (FGFs), which maintain stem cell populations for spermatogenesis [35]. All these factors and others not yet identified could contribute to the promotion and maintenance of spermatogenesis and would be necessary for efficient spermatogenesis *in vitro*. It was even reported recently that bone marrow-derived mesenchymal cells co-cultured with testis tissue promoted *in vitro* spermatogenesis, suggesting that an optimal culture condition for spermatogenesis could be achieved via such a paracrine effect from cells from an extra-testicular source [36].

A second possibility is the paracrine effect between STs. Each ST should secrete various molecules; most of these would be metabolites, and some of these metabolites could directly or indirectly promote spermatogenesis. In organ culture experiments, a testicular tissue mass with a certain volume may be able to create its own internal state through such a paracrine effect. This issue, i.e., the difference between isolated and aggregated STs, could be investigated experimentally in the future by improving the culture technique for isolated STs. In line with such research progress, trials culturing reaggregated testis cells rather than ST segments would be a fruitful technique for *in vitro* spermatogenesis [10, 11].

We expect that the isolated ST culture method would open a new platform for precision culture experiments, especially when combined with microfluidic technologies. It has been reported that microfluidic systems can produce a continuous microflow of culture medium to produce a desired concentration gradient of a biochemical [37, 38]. Combined with such technologies, isolated ST culture experiments could reveal the regulatory mechanisms of spermatogenesis, which have been difficult to elucidate to date.

## Conclusion

Spermatogenesis does not proceed by germ cells alone, but requires close support from several types of testicular somatic cells. This suggests that an organ culture method would be the best approach for *in vitro* spermatogenesis. However, in order to accurately elucidate the regulatory mechanisms of spermatogenesis, it is advantageous and preferable to culture only a small number of functional units—or one functional unit, if possible—rather than large tissue masses. In this study, we showed that *in vitro* spermatogenesis was possible in an isolated ST up to the elongating spermatid stage. There are certainly many limitations and unanswered questions in regard to this technique, including technical difficulty in handling STs, low efficiency of spermatogenesis, unknown effects of interstitial cells, fertility competence of the elongating spermatids, and so forth. Nonetheless, this method allows us to evaluate various microenvironmental parameters involved in the progression of spermatogenesis and to elucidate the regulatory mechanisms of spermatogenesis.

## Supporting information

**S1 File. Six figures and a table.**
(PDF)

**S1 Video. Procedure of the soft agarose method.**
(MP4)

## Acknowledgments

We thank Mayuka Nishida and Shino Nagata for creating the figure illustrations.

## Author Contributions

**Conceptualization:** Takehiko Ogawa.

**Data curation:** Xuemin Feng.

**Formal analysis:** Takafumi Matsumura.

**Investigation:** Xuemin Feng, Takafumi Matsumura, Yuki Yamashita, Kiyoshi Hashimoto, Hisakazu Odaka.

**Project administration:** Takehiko Ogawa.

**Resources:** Yoshinori Makino, Yuki Okada, Hiroko Nakamura, Hiroshi Kimura.

**Supervision:** Takuya Sato, Teruo Fujii.

**Writing – original draft:** Xuemin Feng.

**Writing – review & editing:** Takehiko Ogawa.

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
