## [Decision Letter · Decision Letter 0]

15 Aug 2022

PONE-D-22-19509Mouse in vitro spermatogenesis in isolated seminiferous tubulesPLOS ONE

Dear Dr. Ogawa,

Thank you for submitting your manuscript to PLOS ONE. After careful consideration, we feel that it has merit but does not fully meet PLOS ONE’s publication criteria as it currently stands. Therefore, we invite you to submit a revised version of the manuscript that addresses the points raised during the review process. There are some concerns on the efficiency and extent of spermatogenesis and culturing the tissues in environment that lack the factors that are crucial in an in vivo system. Further, there are a number of issues with methodology. Presentation of the manuscript should be improved significantly, especially in providing strong rationale and citing up to date literature; clear mention of the methodologies used; statistical analyses; appropriately describing the results for easy understanding; and providing the discussion that is relevant to the results observed backed by strong literature survey.

We look forward to receiving your revised manuscript.

Kind regards,

Suresh Yenugu

Academic Editor

PLOS ONE

Journal Requirements:

Reviewers' comments:

Reviewer's Responses to Questions

**Comments to the Author**

1. Is the manuscript technically sound, and do the data support the conclusions?

Reviewer #1: Partly

Reviewer #2: No

Reviewer #3: Yes

2. Has the statistical analysis been performed appropriately and rigorously? 

Reviewer #1: Yes

Reviewer #2: I Don't Know

Reviewer #3: Yes

3. Have the authors made all data underlying the findings in their manuscript fully available?

Reviewer #1: Yes

Reviewer #2: No

Reviewer #3: Yes

4. Is the manuscript presented in an intelligible fashion and written in standard English?

Reviewer #1: Yes

Reviewer #2: No

Reviewer #3: Yes

5. Review Comments to the Author

Reviewer #1: Comments:

Recapitulation of in vitro spermatogenesis is essential to study the molecular mechanisms and to address infertility. Feng et al. extended their previous studies on in vitro spermatogenesis using the ST culture method using Acr-GFP andH3 mice. The authors tried both minced or individual STs and aggregated STs. Interestingly, they found that aggregates or individual STs having aggregates at both ends show efficient spermatogenesis compared to individual STs. Further, using ST culturing, authors have shown the effect of oxygen concentration on spermatogenesis. This is a motivating extension work; however, some aspects of the study need to be clarified, and more experiments are necessary to describe the manuscript's findings. Also, serious issues with the writing must be addressed. Overall, the paper feels premature and would benefit from some major rewriting. Throughout the article, including the abstract, many vocabularies, grammar, and pose errors make it very hard to follow and interpret.

Major comments:

The abstract is poorly written.

If possible, I would suggest to the authors consider making the video of methods A and B. It will be beneficial to the researchers and readers.

Authors made major conclusions that spermatogenesis can be induced in short pieces of STs in culture. However, only 26% of the short pieces show the +ve signals of GFP and mcherry. Also, they are nearby small clustered regions. Of course, we can see the GFP signals but Mcherry and GFP colocalized signals we see nearby clusters only?

Here the point is with the low efficiency of spermatogenesis in short pieces, how come one can use this technique to study mechanisms of spermatogenesis or any toxicological studies?

Line 465: Numerous studies have described the pivotal roles of these cells in spermatogenesis. Leydig cells produce and secrete androgens and other factors, such as colony-stimulating factor 1 (CSF-1) [31]. The testicular macrophages also produce CSF-1 and enzymes involved in retinoic acid (RA) biosynthesis [32]. Lymphatic endothelial cells secrete fibroblast growth factors (FGFs), which maintain stem cell populations for spermatogenesis [33]. All these factors and others not yet identified could contribute to the promotion and maintenance of spermatogenesis and would be necessary for efficient spermatogenesis in vitro.

As the authors stated ST’s external microenvironment may play a critical role in spermatogenesis. Did the authors try to culture short pieces of STs with some of the factors like RA, CSF-1 or growth factors? In another way did the authors try to co-culture with Leydig cells or other cells required for spermatogenesis to improve the efficiency?

Minor comments:

Fig1G Y axis?

323 expression was observed in 55 out of 90 sa268mples (61.1%) (Fig 3D, E).

Fig 3G ; error bars labelled. Make it uniform or mention it in legends

Fig S1 and 2 methods 0.5% agarose solutions with STs kept in 4deg. How about cold stress or shock on tubules and spermatocytes?

Fig3: What % of aggregates were observed? What do the authors mean occasionally?

Line 327: In 40–55 days, STs from an experiment were examined by whole mount immunochemical staining. In 2 samples of cut-isolated ST, cells positive for both GFP and mCherry were observed out of 41 samples examined

The image shown +ve for GPF and Mcherry in cut isolate is not convincing. Because the +ve signals we see near the cluster.

Line 334: We then stained Leydig cells with anti-HSD3β antibody under whole mount condition. In fact, HSD3β-positive cells were detected abundantly in aggregated STs and only marginally in isolated ST samples (Fig 3I). Please add the quantification

Line 340: In our previous study using rat testis tissues, we found that O2 tension could influence the spermatogenic efficiency in vitro and O2 concentration lower than atmospheric 20% would be favourable : Rewrite

377 that this condition, namely uncut-isolated ST under 10%O2: 10% space

Overlapping scale bars

Make uniformity in scale bars

Fig2:

(B) Even though authors have mentioned that Acr-GFP expression in an ST showed unique oscillatory fluctuation, the authors need to explain the reason for the sharp fluctuation in Acr-GFP expression from Day 24 to Day 31.

Fig3:

(F) The authors examined only 41 cut-isolated STs for both GFP –mCherry expression. What about the rest of the cut-isolated STs as the authors have found quite significant (72 nos) GFP expressing cut-isolated STs?

In lines 318, 319 and 320, they found small clumps of STs which they have considered aggregated STs. The authors need to clarify whether this is a human error or if these clumps are forming on their own. In the latter case, the authors also need to mention the frequency and percentage of finding these clumps.

In addition to this, the authors need to clarify whether they have included these clumped STs in their count for aggregated STs or not.

(I) The authors need to provide statistical analysis for this figure as they have stated that ‘aggregated STs may maintain interstitial cells between each ST, which could be a reason that aggregated STs supported spermatogenesis more favourably.’

Fig. 4

(B) Out of 20%, 15% and 10% concentrations, in 10% O2 GFP expression is more. Whether this is the optimal concentration for spermatogenesis or there is the possibility of more spermatogenesis below 10%? The author needs to clarify.

Reviewer #2: In this study the authors evaluated the effect of oxygen concentration on in vitro spermatogenesis in neonatal Acr-GFP/Histone H3.3.3-mCherry double (Acr/H3) transgenic mice testes, using tissue mass, ST aggregate and ST segment culture systems. Although it seems to be a have short follow up duration s (34 day long), the authors report that the lower oxygen tension is favorable for spermatogenesis and for elongating spermatid production, and they demonstrated a new ST segment culture method providing opportunity for elucidating regulatory mechanism of spermatogenesis. While these data are potentially of interest, the manuscript lacks principal points that require a major irevision in order to ensure the data are presented with more clarity and greater empirical support (including controls and detailed data).

Title:could be intentional in order to correctly orient the readers to the output of the study.

Introduction: lacks current literature and it is written in a broad and unfocused manner with a book chapter style. Therefore the rationale (lack of knowledge in recent literature related to the research question) , the clear description of the problem and the hypothesis or the research question that will solve the problem should be clearly declared .

Materials and Method lacks the study design, the sample size with power analysis, the number of repeats taht is required in order to answer the research question objectively. Several concerns related to methodolody and the proff of concepts are as following.

1. Please provide the detailed information of PDMS ceiling chip which was used for seminiferous tubule culturing method (including size of pillars in WxLxH and distance between the pillars) in M&M section.

2. In M&M section, between lines 121-124, the usage of a thin porous polycarbonate membrane is stated in some cases for pressing testis tissues against the base agarose gel by placing them under the PDMS ceiling to hold the tissue in place. Please clarify the optimization of culture systems by explaining in which cases the membrane was used.

3. The all culture platforms used in study should be clarified in M&M section. Please change the title "Culture method" as "Culture methods" in line 136 and explain each culture methods as subtitles individually. The title of "Soft agarose method" in line 156 should also be under culture methods title as a subtitle.

4. In figures 2A and 2B, there is high background for GFP, that makes the labeling data very unreliable.

5. The terminology for culture systems is not consistent. Each culture system should be named in M&M section and the same names should be used in the remaining parts of the manuscript.

6. In figure 4F, PAS staining micrograph is given only for uncut-isolated ST culture, it should be also given for the other culture platforms for the same culture duration. The oxygen concentration is also should be provided in figure legend.

7. It seems that there are four culture methods as (1) cut-isolated ST (PDMS ceiling), (2) uncut-isolated ST (PDMS ceiling with pillars), (3) ST aggregate (soft agarose), (4) tissue mass culture (air-liquid interphase). In results section, the comparison of these 4 methods should be provided (in figure 1G, 3E, 3H, 4B and 4H). In figure 1G, it should be stated which oxygen concentration is used and what is the culture time for that analysis.

The results lack the answers of the research questions and Discussion part itatement of limitations of the current study. The changes in metabolites (analysis of culture media by LC-MS etc.), the functionality of elongated spermatids in terms of fertilization (ROSI, ICSI) or genetic stability tests should be performed or may be added as limitations.

In discussion section, there are missing references in terms of comparison of the efficiency of new cut-isolated ST and uncut-isolated ST culture systems. The following articles should be discussed in terms of in vitro spermatogenetic process:

- Önen S, Köse S, Yersal N, Korkusuz P. Mesenchymal stem cells promote spermatogonial stem/progenitor cell pool and spermatogenesis in neonatal mice in vitro. Sci Rep. 2022 Jul 7;12(1):11494. doi: 10.1038/s41598-022-15358-5. PMID: 35798781; PMCID: PMC9263145.

- Baert Y, Dvorakova-Hortova K, Margaryan H, Goossens E. Mouse in vitro spermatogenesis on alginate-based 3D bioprinted scaffolds. Biofabrication. 2019 Apr 26;11(3):035011. doi: 10.1088/1758-5090/ab1452. PMID: 30921781.

- AbuMadighem A, Shuchat S, Lunenfeld E, Yossifon G, Huleihel M. Testis on a chip-a microfluidic three-dimensional culture system for the development of spermatogenesisin-vitro. Biofabrication. 2022 Apr 20;14(3). doi: 10.1088/1758-5090/ac6126. PMID: 35334473.

Reviewer #3: In this study, the authors developed an optimized method to culture the segment of seminiferous tubule (ST), and they showed that ST culture could reach the elongating spermatids stage. The authors patiently troubleshot the strategies to culture ST. They carefully described the detailed processes of their trial and error to develop an optimized protocol. I found the study valuable because the authors describe how they overcame the technical difficulties. This study would become an important asset in the field. Readers can understand the key process of their optimization. Further, the study demonstrates the impact of oxygen concentration in culture and clarifies the benefit of a hypoxic condition.

1. Some sentences in the abstract sound awkward. Lines 27-19 "Theoretically,”, and the last sentence in Lines 32-33.

2. Supplementary figures can be organized in the order of descriptions in the result section. S2A can be S1.

3. Each result section consists of a single paragraph. These sections can be reorganized into some paragraphs to improve their readability.

4. In Fig.1F, it was shown that the frequencies of GFP-positive cells were variable, but these data are not clearly shown in Figure G. These two data can be better described to clarify the frequency of GFP-positive cells.

5. Fig. S2: Please add labels: Acr-GFP and H3.3-mCherry.

6. Line 316: mislabel “S1 Fig. Method B”.

7. Line 334-336: This is just a comment. Leydig cells are the primary source of testosterone or androgens in males. Testosterone or androgens can be supplemented in future studies.

6. PLOS authors have the option to publish the peer review history of their article (what does this mean?). If published, this will include your full peer review and any attached files.

Reviewer #1: No

Reviewer #2: No

Reviewer #3: No

---

## [Author Response · Author response to Decision Letter 0]

18 Oct 2022

Reviewer #1: Comments:

Recapitulation of in vitro spermatogenesis is essential to study the molecular mechanisms and to address infertility. Feng et al. extended their previous studies on in vitro spermatogenesis using the ST culture method using Acr-GFP andH3 mice. The authors tried both minced or individual STs and aggregated STs. Interestingly, they found that aggregates or individual STs having aggregates at both ends show efficient spermatogenesis compared to individual STs. Further, using ST culturing, authors have shown the effect of oxygen concentration on spermatogenesis. This is a motivating extension work; however, some aspects of the study need to be clarified, and more experiments are necessary to describe the manuscript's findings. Also, serious issues with the writing must be addressed. Overall, the paper feels premature and would benefit from some major rewriting. Throughout the article, including the abstract, many vocabularies, grammar, and pose errors make it very hard to follow and interpret.

Major comments:

The abstract is poorly written.

Reply) I have rewritten the abstract to be more specific about what we did in this study. I think that the new abstract is more informative and easier to understand for most readers.

If possible, I would suggest to the authors consider making the video of methods A and B. It will be beneficial to the researchers and readers.

Reply) We have made a video showing the procedure of soft agarose methods A and B, which was attached as supplementary data. Thank you for your suggestion.

Authors made major conclusions that spermatogenesis can be induced in short pieces of STs in culture. However, only 26% of the short pieces show the +ve signals of GFP and mcherry. Also, they are nearby small clustered regions. Of course, we can see the GFP signals but Mcherry and GFP colocalized signals we see nearby clusters only?

Reply) Reading your comment, I came to notice that GFP- and mCherry-double positive cells were observed at locations in a tubule which was not stretched but bent or twisted to form ST aggregation-like configuration. This may have an important implication for a future study. In any case, however, they were present in each singly cut-isolated ST, but not in STs of clustered or aggregated. 

Here the point is with the low efficiency of spermatogenesis in short pieces, how come one can use this technique to study mechanisms of spermatogenesis or any toxicological studies?

Reply) The Acr-GFP positivity rate of isolated STs is indeed lower than that of ST aggregates or tissue mass cultures. However, I consider this as follows. In order to improve culture condition for spermatogenesis, we have to test variety of interventions, including supplements in the medium and physical conditions like different O2 concentration. Currently, the regular testicular tissue culture method, namely tissue mass culture, has achieved an efficiency as high as 90% Acr-GFP positivity. With this system, it may be difficult to find new culture conditions that are better than the control condition. For example, even if the addition of substance X to the culture medium is effective in promoting spermatogonial proliferation and inducing spermatogenesis, the effect of substance X may be overlooked in experiments where there is little room for improvement, with Acr-GFP expression rates as high as 90% in the control group. On the contrary, in this context, low spermatogenic efficiency of the isolated ST culture method may be useful to find better culture conditions. In fact, effect of O2 concentration was examined vividly in the present study to show lower O2 is better than 20%. 

Line 465: Numerous studies have described the pivotal roles of these cells in spermatogenesis. Leydig cells produce and secrete androgens and other factors, such as colony-stimulating factor 1 (CSF-1) [31]. The testicular macrophages also produce CSF-1 and enzymes involved in retinoic acid (RA) biosynthesis [32]. Lymphatic endothelial cells secrete fibroblast growth factors (FGFs), which maintain stem cell populations for spermatogenesis [33]. All these factors and others not yet identified could contribute to the promotion and maintenance of spermatogenesis and would be necessary for efficient spermatogenesis in vitro.

As the authors stated ST’s external microenvironment may play a critical role in spermatogenesis. Did the authors try to culture short pieces of STs with some of the factors like RA, CSF-1 or growth factors? In another way did the authors try to co-culture with Leydig cells or other cells required for spermatogenesis to improve the efficiency?

Reply) Experiments suggested here are all important trial in our next studies. We will certainly plan such studies with the isolated ST experiment. Another reviewer, Reviewer 2, suggested to read a very recent paper by Önen et al. which used transwell insert system to coculture bone marrow derived mesenchymal cells and the testis tissue. Such co-culture system can be used to test your suggestion. 

Minor comments:

Fig1G Y axis?

Reply) The Y-axis represents the percentage of GFP-positive samples out of all samples cultured. The title of the Y-axis was added to Fig. 1G of the revised manuscript.

323 expression was observed in 55 out of 90 sa268mples (61.1%) (Fig 3D, E).

Reply) I’m sorry for this careless typing mistake. I have corrected it in the revised manuscript. 

Fig 3G; error bars labelled. Make it uniform or mention it in legends

Reply) In revision, I have erased them to make it uniform in all figures.

Fig S1 and 2 methods 0.5% agarose solutions with STs kept in 4deg. How about cold stress or shock on tubules and spermatocytes?

Reply) In our experience, storage at 4°C for 15 minutes does not affect the culture results.

Fig3: What % of aggregates were observed? What do the authors mean occasionally?

Reply) Thank you for this comment. Now I think “occasionally” was an incorrect word to describe the case. It totally depends on how rigorously the practitioner disassembled seminiferous tubules from each other. It is possible to eliminate aggregates completely, but it takes time and labor. Thus, in practice, ST aggregates remained more or less in every case, to varying degrees. 

Line 327: In 40–55 days, STs from an experiment were examined by whole mount immunochemical staining. In 2 samples of cut-isolated ST, cells positive for both GFP and mCherry were observed out of 41 samples examined

The image shown +ve for GPF and Mcherry in cut isolate is not convincing. Because the +ve signals we see near the cluster.

Reply) As I replied above, the sample was a singly cut-isolated ST in the soft agarose. The “cluster” you mentioned is supposed to be the result of ST being bent or twisted during the whole mount immunohistochemistry procedure. 

Line 334: We then stained Leydig cells with anti-HSD3β antibody under whole mount condition. In fact, HSD3β-positive cells were detected abundantly in aggregated STs and only marginally in isolated ST samples (Fig 3I). Please add the quantification

Reply) I counted the number of HSD3β-positive cells in the picture. The data were described in the figure legend as follows. 

“HSD3β signal was observed in cells adhered to the outside of the STs, either singly (arrows) or as clusters (arrow heads). In these pictures, such HSD3β-positive cells were counted to be 5 and 41 in cut-isolated and aggregated STs, respectively. In addition, in case of aggregated STs, a clumpy interstitial tissue containing about 20 HSD3β-positive cells was also observed (surrounded by braces).”

Line 340: In our previous study using rat testis tissues, we found that O2 tension could influence the spermatogenic efficiency in vitro and O2 concentration lower than atmospheric 20% would be favourable : Rewrite

Reply) I rewrote as below in the revised manuscript. 

In our previous study using rat testis tissues, we found that an O2 concentration lower than atmospheric concentration of 20% induced spermatogenesis with higher efficiency [20].

377 that this condition, namely uncut-isolated ST under 10%O2: 10% space

Reply) Thank you for this comment. I placed a space at that position in the revised manuscript. 

Overlapping scale bars

Make uniformity in scale bars

Reply) Scale bars were examined again and corrected. Thank you.

Fig2:

(B) Even though authors have mentioned that Acr-GFP expression in an ST showed unique oscillatory fluctuation, the authors need to explain the reason for the sharp fluctuation in Acr-GFP expression from Day 24 to Day 31.

Reply) Yes, it was surprising to see such a drastic change in GFP expression in a single ST in 7 days. I consider this phenomenon as follows. The Acr-GFP expression starts at stage Ⅳ pachytene spermatocyte. Therefore, STs on day 24 in the figure should have contained germ cells of stage Ⅳ spermatocyte and some more differentiated cells. However, those GFP-expressing germ cells were about to die and disappeared in next 4 days. On day 28, therefore, the ST became GFP-negative, containing only germ cells of stage Ⅲ spermatocyte and less differentiated germ cells. Over the next 3 days, the stage Ⅲ spermatocytes turned into stage Ⅳ spermatocyte and beyond, and on day 31, the ST became GFP-positive again. 

Fig3:

(F) The authors examined only 41 cut-isolated STs for both GFP –mCherry expression. What about the rest of the cut-isolated STs as the authors have found quite significant (72 nos) GFP expressing cut-isolated STs?

Reply) We have not examined the rest 31 samples for the mCherry expression.

In lines 318, 319 and 320, they found small clumps of STs which they have considered aggregated STs. The authors need to clarify whether this is a human error or if these clumps are forming on their own. In the latter case, the authors also need to mention the frequency and percentage of finding these clumps.

Reply) Yes, the clump STs was named as “aggregated STs”. STs do not self-aggregate in the soft agarose. They were results of incomplete dissociation of STs by the practitioner. 

In addition to this, the authors need to clarify whether they have included these clumped STs in their count for aggregated STs or not.

Reply) Yes, clumped STs were counted as aggregated ST. 

(I) The authors need to provide statistical analysis for this figure as they have stated that ‘aggregated STs may maintain interstitial cells between each ST, which could be a reason that aggregated STs supported spermatogenesis more favourably.’

Reply) Aggregated STs are a kind of artifact, unintentionally generated in soft agarose experiments. 

The characteristics of aggregated STs, i.e., size, density, number of stromal cells, etc., may vary widely among them. For this reason, we hesitated to examine them in any further detail, after whole mount immunohistochemistry was performed once. We merely proposed that a greater number of interstitial cells may be responsible for more efficient spermatogenesis in aggregated STs. Thus, this is not definitive nor the conclusion of this study. I made it clearer in the revised manuscript. 

Fig. 4

(B) Out of 20%, 15% and 10% concentrations, in 10% O2 GFP expression is more. Whether this is the optimal concentration for spermatogenesis or there is the possibility of more spermatogenesis below 10%? The author needs to clarify.

Reply) I think it is possible that a particular O2 concentration which is lower than 10% could be optimal for spermatogenesis in single ST. From our present study, it seems a delicate issue and the optimal O2 concentration may change in each tissue depending on the tissue size (O2 consumption rate). I wrote this in the revised manuscript. 

Reviewer #2: In this study the authors evaluated the effect of oxygen concentration on in vitro spermatogenesis in neonatal Acr-GFP/Histone H3.3.3-mCherry double (Acr/H3) transgenic mice testes, using tissue mass, ST aggregate and ST segment culture systems. Although it seems to be a have short follow up duration s (34 day long), the authors report that the lower oxygen tension is favorable for spermatogenesis and for elongating spermatid production, and they demonstrated a new ST segment culture method providing opportunity for elucidating regulatory mechanism of spermatogenesis. While these data are potentially of interest, the manuscript lacks principal points that require a major irevision in order to ensure the data are presented with more clarity and greater empirical support (including controls and detailed data).

Title:could be intentional in order to correctly orient the readers to the output of the study.

Reply) I interpret this comment to mean that the current title is ambiguous and does not include the conclusions of this study. So, I considered other titles. 

Present title; “Mouse in vitro spermatogenesis in isolated seminiferous tubules”. 

New title 1; “Mouse spermatogenesis can proceed in isolated seminiferous tubules”.

New title 2; “Spermatogenesis proceeded in isolated seminiferous tubules of immature mice”. 

New title 3; “???”

Thinking again, I reached the original one would be the best. Yet, I’m open to suggestions.

Introduction: lacks current literature and it is written in a broad and unfocused manner with a book chapter style. Therefore the rationale (lack of knowledge in recent literature related to the research question) , the clear description of the problem and the hypothesis or the research question that will solve the problem should be clearly declared .

Reply) I rewrote the introduction which included recent literatures. As for the research question, we focus on a very simple question which is whether a singly isolated ST can induce and support spermatogenesis or not. The answer was turned out to be yes. To make the introduction to be compact, we omitted the description of double Tg mouse. 

Materials and Method lacks the study design, the sample size with power analysis, the number of repeats taht is required in order to answer the research question objectively. 

Reply) The Materials and Methods section was reorganized. In addition, to facilitate the understanding of the present study as a whole, I have made a table along with graph encompassing every method together and present it as S6 Fig.

Several concerns related to methodolody and the proff of concepts are as following.

1. Please provide the detailed information of PDMS ceiling chip which was used for seminiferous tubule culturing method (including size of pillars in WxLxH and distance between the pillars) in M&M section.

Reply) I added that information in the M&M section. 

2. In M&M section, between lines 121-124, the usage of a thin porous polycarbonate membrane is stated in some cases for pressing testis tissues against the base agarose gel by placing them under the PDMS ceiling to hold the tissue in place. Please clarify the optimization of culture systems by explaining in which cases the membrane was used.

Reply) First of all, I made a mistake to have written that polycarbonate membrane was used to press tissue mass. It was not polycarbonate membrane but a PDMS membrane (75 μm thick, FS7075C, ASAHI RUBBER inc). So, it was corrected accordingly in the revised manuscript. 

This procedure was not relevant to the optimization of culture condition. The membranes were placed on the tissue mass but not on the isolated part of the seminiferous tubule. Therefore, we do not think it affect the culture results, nor relating to the optimization of culture condition. 

3. The all culture platforms used in study should be clarified in M&M section. Please change the title "Culture method" as "Culture methods" in line 136 and explain each culture methods as subtitles individually. The title of "Soft agarose method" in line 156 should also be under culture methods title as a subtitle.

Reply) I reorganized the M&M section. “Soft agarose method” was changed to “Culture methods - Soft agarose method” and a new section of “Culture methods – PDMS ceiling (PC) method” was added to explain the detail of uncut-isolated ST method.

4. In figures 2A and 2B, there is high background for GFP, that makes the labeling data very unreliable.

Reply) That is the best we can do now with our microscope and camera setting. Sorry.

5. The terminology for culture systems is not consistent. Each culture system should be named in M&M section and the same names should be used in the remaining parts of the manuscript.

Reply) I made their name consistent, which was summarized in S6 Fig. 

6. In figure 4F, PAS staining micrograph is given only for uncut-isolated ST culture, it should be also given for the other culture platforms for the same culture duration. The oxygen concentration is also should be provided in figure legend.

Reply) In this study, we took Acr-GFP and H3.3-mCherry as the reliable parameter for evaluating spermatogenic progression. I think they are basically enough to support our conclusion. Immunohistochemical staining and histological PAS stain were also performed to confirm and reinforce the conclusion. That is the reason that PAS staining was only performed and shown in the final figure. The oxygen concentration 10% was provided in the figure legend of Fig. 4E & F. 

7. It seems that there are four culture methods as (1) cut-isolated ST (PDMS ceiling), (2) uncut-isolated ST (PDMS ceiling with pillars), (3) ST aggregate (soft agarose), (4) tissue mass culture (air-liquid interphase). In results section, the comparison of these 4 methods should be provided (in figure 1G, 3E, 3H, 4B and 4H). In figure 1G, it should be stated which oxygen concentration is used and what is the culture time for that analysis.

Reply) I appreciate your attentive reading of our manuscript. Present our research may not be organized in an ordinary style, because I described not only the final form of our best method but also methods we tested in our endeavor to improve them. I thank you for giving me an opportunity to think it again for summarizing and comparing culture methods. 

Now I summarized our method as follows, which was presented in a table in S6 Fig. 

(1) cut-isolated conglomerated ST (PDMS ceiling) 

(2) uncut-isolated ST (PDMS ceiling with pillars) 

(3) tissue mass culture (PDMS ceiling)

(4) cut-isolated ST in soft agarose

(5) ST aggregate in soft agarose

As for Fig. 1G, the culture time for the analysis was variable indeed. Each experiment has different culture duration from 35 to 77 days. The Acr-GFP positive sign (>10% area) was taken whenever during the culture period. I have added this information in the figure legend.

The results lack the answers of the research questions and Discussion part itatement of limitations of the current study. The changes in metabolites (analysis of culture media by LC-MS etc.), the functionality of elongated spermatids in terms of fertilization (ROSI, ICSI) or genetic stability tests should be performed or may be added as limitations.

Reply) Yes, there are many open questions in this study. We started this study asking whether isolated ST can induce and support spermatogenesis or not. To this question, we answered yes, it is possible to culture isolated ST for inducing spermatogenesis, although with lowered efficiency than tissue mass culture. I added sentences about the limitation of this study in the Conclusion of revised manuscript. 

In discussion section, there are missing references in terms of comparison of the efficiency of new cut-isolated ST and uncut-isolated ST culture systems. The following articles should be discussed in terms of in vitro spermatogenetic process:

- Önen S, Köse S, Yersal N, Korkusuz P. Mesenchymal stem cells promote spermatogonial stem/progenitor cell pool and spermatogenesis in neonatal mice in vitro. Sci Rep. 2022 Jul 7;12(1):11494. doi: 10.1038/s41598-022-15358-5. PMID: 35798781; PMCID: PMC9263145.

Reply) Thank you recommending this paper to read. It is really interesting that bone marrow derived mesenchymal stem cells cultured on the well-bottom have promotive effect on in vitro spermatogenesis of testis tissue placed above in the transwell insert. This may mean that same method can improve the efficiency of spermatogenesis in the ST culture experiment. I added a sentence with this reference in the discussion. 

- Baert Y, Dvorakova-Hortova K, Margaryan H, Goossens E. Mouse in vitro spermatogenesis on alginate-based 3D bioprinted scaffolds. Biofabrication. 2019 Apr 26;11(3):035011. doi: 10.1088/1758-5090/ab1452. PMID: 30921781.

Reply) This paper is also interesting showing that testis cells once singly dispersed were reaggregated by bioprinting method, in which spermatogenic progression was observed in following cultivation. Thus, this study may indicate a future direction of our present study too. 

- AbuMadighem A, Shuchat S, Lunenfeld E, Yossifon G, Huleihel M. Testis on a chip-a microfluidic three-dimensional culture system for the development of spermatogenesis in-vitro. Biofabrication. 2022 Apr 20;14(3). doi: 10.1088/1758-5090/ac6126. PMID: 35334473.

Reply) As above paper, this paper also showed culturing reaggregated testis cell mass in a microfluidic device. I included this and above paper together in the discussion. 

Reviewer #3: In this study, the authors developed an optimized method to culture the segment of seminiferous tubule (ST), and they showed that ST culture could reach the elongating spermatids stage. The authors patiently troubleshot the strategies to culture ST. They carefully described the detailed processes of their trial and error to develop an optimized protocol. I found the study valuable because the authors describe how they overcame the technical difficulties. This study would become an important asset in the field. Readers can understand the key process of their optimization. Further, the study demonstrates the impact of oxygen concentration in culture and clarifies the benefit of a hypoxic condition.

1. Some sentences in the abstract sound awkward. Lines 27-19 "Theoretically,”, and the last sentence in Lines 32-33.

Reply) I appreciate your comments. I have rewritten the abstract and those sentences were removed. Thank you.

2. Supplementary figures can be organized in the order of descriptions in the result section. S2A can be S1.

Reply) The numbers were reordered in the revised version.

3. Each result section consists of a single paragraph. These sections can be reorganized into some paragraphs to improve their readability.

Reply) I do appreciate this suggestion. In the revision, the section before the last, Cultivation of cut-isolated STs using a soft agarose method, was divided into 3 paragraphs. The last section, Lowered oxygen concentration was favorable for spermatogenesis, was divided into 5 paragraphs.

4. In Fig.1F, it was shown that the frequencies of GFP-positive cells were variable, but these data are not clearly shown in Figure G. These two data can be better described to clarify the frequency of GFP-positive cells.

Reply) Thanks for this comment. To make the difference among three groups, I made two thresholds, 10% and 50% of GFP-positive area, under observation with the stereomicroscope. The graph in figure 1G was modified to show GFP-positive rates with both of these two thresholds. 

5. Fig. S2: Please add labels: Acr-GFP and H3.3-mCherry.

Reply) The labels were changed accordingly. Thanks.

6. Line 316: mislabel “S1 Fig. Method B”.

Reply) Sorry for this careless mistake. It was corrected in the revised manuscript.

7. Line 334-336: This is just a comment. Leydig cells are the primary source of testosterone or androgens in males. Testosterone or androgens can be supplemented in future studies.

Reply) Yes, I think the supplementation of testosterone and other substances secreted by Leydig cells should be done in next experiment. Thank you.

---

## [Decision Letter · Decision Letter 1]

15 Nov 2022

PONE-D-22-19509R1Mouse in vitro spermatogenesis in isolated seminiferous tubulesPLOS ONE

Dear Dr. Ogawa,

Thank you for submitting your manuscript to PLOS ONE. After careful consideration, we feel that it has merit but does not fully meet PLOS ONE’s publication criteria as it currently stands. Therefore, we invite you to submit a revised version of the manuscript that addresses the points raised during the review process. While major improvements are made to the manuscript, it is suggested that the authors provide control data for each and every time point assessed. Ensure that the comments of the reviewers are responded in detail. The manuscript should be corrected for English language.

We look forward to receiving your revised manuscript.

Kind regards,

Suresh Yenugu

Academic Editor

PLOS ONE

Reviewers' comments:

Reviewer's Responses to Questions

**Comments to the Author**

1. If the authors have adequately addressed your comments raised in a previous round of review and you feel that this manuscript is now acceptable for publication, you may indicate that here to bypass the “Comments to the Author” section, enter your conflict of interest statement in the “Confidential to Editor” section, and submit your "Accept" recommendation.

Reviewer #1: All comments have been addressed

Reviewer #2: (No Response)

Reviewer #3: (No Response)

2. Is the manuscript technically sound, and do the data support the conclusions?

Reviewer #1: Yes

Reviewer #2: No

Reviewer #3: Yes

3. Has the statistical analysis been performed appropriately and rigorously? 

Reviewer #1: Yes

Reviewer #2: No

Reviewer #3: Yes

4. Have the authors made all data underlying the findings in their manuscript fully available?

Reviewer #1: Yes

Reviewer #2: No

Reviewer #3: Yes

5. Is the manuscript presented in an intelligible fashion and written in standard English?

Reviewer #1: Yes

Reviewer #2: No

Reviewer #3: Yes

6. Review Comments to the Author

Reviewer #1: The authors have done a good job revising their manuscript, which is clearly improved and ready for acceptance. Overall, the authors addressed most of my previous comments.

Reviewer #2: The revised manuscript has been improved to some extent, in terms of quality with reorganized Abstract, Introduction, M&M sections and terminology for culture methods. On the other hand, the reliability of data in terms of time points, oxygen concentrations and sample size are not clear. Following are the comments.

1) The authors might provide a more appropriate title for the manuscript including the culture method used for isolated seminiferous tubules that constitutes the main key word of the study. In the present title, “Mouse in vitro spermatogenesis in isolated seminiferous tubules”, the term of “isolated seminiferous tubules” might point the isolation of seminiferous tubules from an adult mouse. Also, the age of the mouse and effect of oxygen level should be indicated in the title. Here you can find some options as possible titles:

2) The authors are strictly recommended to provide a more reliable data for comparison of groups at same time points in figures 1G, 3E, 3H and 4B. For example, in figure 1G, if the culture time for the analysis is variable from 35 to 77 days, it is not possible to compare the efficacy of cut-isolated ST, uncut-isolated ST and tissue mass culture methods in terms of GFP expression since the progression of the spermatogenesis is changeable in in vitro conditions.The time point, sample size (n=?) and oxygen concentration must be given in legends of figures 1G, 3E, 3H and 4B.

3) All of the comments should be answered one by one on the rebuttal letter; then should address to the revised manuscript's specified lines/paragraphs/pages where the question is clearly answered.

4) The language that is used both on the rebuttal letter and the manuscript definitely requires a major refinement and revision in terms of scientific approach, the medical terminology and grammar.

Reviewer #3: The authors fully addressed my previous comments. It became a solid study. One minor thing I found is that the abstract contains too much details now, I feel. It can be reorganized in a way readers can capture the big picture.

7. PLOS authors have the option to publish the peer review history of their article (what does this mean?). If published, this will include your full peer review and any attached files.

Reviewer #1: No

Reviewer #2: No

Reviewer #3: No

---

## [Author Response · Author response to Decision Letter 1]

13 Dec 2022

6. Review Comments to the Author

Reviewer #1: The authors have done a good job revising their manuscript, which is clearly improved and ready for acceptance. Overall, the authors addressed most of my previous comments.

Reply: I appreciate your kind comments and suggestions.

Reviewer #2: The revised manuscript has been improved to some extent, in terms of quality with reorganized Abstract, Introduction, M&M sections and terminology for culture methods. On the other hand, the reliability of data in terms of time points, oxygen concentrations and sample size are not clear. Following are the comments.

Reply: Thank you for your comments. 

1) The authors might provide a more appropriate title for the manuscript including the culture method used for isolated seminiferous tubules that constitutes the main key word of the study. In the present title, “Mouse in vitro spermatogenesis in isolated seminiferous tubules”, the term of “isolated seminiferous tubules” might point the isolation of seminiferous tubules from an adult mouse. Also, the age of the mouse and effect of oxygen level should be indicated in the title. Here you can find some options as possible titles:

Reply: I appreciate your suggestion. I changed the title as follows:

“In vitro spermatogenesis in isolated seminiferous tubules of immature mice”

2) The authors are strictly recommended to provide a more reliable data for comparison of groups at same time points in figures 1G, 3E, 3H and 4B. For example, in figure 1G, if the culture time for the analysis is variable from 35 to 77 days, it is not possible to compare the efficacy of cut-isolated ST, uncut-isolated ST and tissue mass culture methods in terms of GFP expression since the progression of the spermatogenesis is changeable in in vitro conditions. The time point, sample size (n=?) and oxygen concentration must be given in legends of figures 1G, 3E, 3H and 4B.

Reply: According to your recommendation, I narrowed the range of dates for the analysis in Figure 1G. The examination range was limited to 10 days, from culture days 31 to 40, and some of the data changed a bit. As for Figure 3E, the dates of analysis were culture days 20 to 25, and this information was added to the figure legend. In the case of Figure 3H, the day of tissue fixation for immunohistochemistry was added to the figure legend. Finally, in Figure 4B, GFP-positivity was assessed over culture days 20 to 30, and mCherry expression was confirmed on day 30 or 36. This information was added to the figure legend. Sample sizes are shown in the graph and an appropriate explanation was added to the legends. The oxygen concentration is now given as 20% in the legends for Figures 1G, 3E, and 3H. 

3) All of the comments should be answered one by one on the rebuttal letter; then should address to the revised manuscript's specified lines/paragraphs/pages where the question is clearly answered.

Reply: Thank you for the advice. 

4) The language that is used both on the rebuttal letter and the manuscript definitely requires a major refinement and revision in terms of scientific approach, the medical terminology and grammar.

Reply: Yes, I wrote the rebuttal letter ourselves and perhaps should have sought editorial help. As for the manuscript, however, I sent it to a professional English-editing company, where it was proofread and corrected by a native-English speaking editor. I sent the current rebuttal letter along with the revised manuscript to the company again. 

Reviewer #3: The authors fully addressed my previous comments. It became a solid study. One minor thing I found is that the abstract contains too much details now, I feel. It can be reorganized in a way readers can capture the big picture.

Reply: Thank you for your kind comments and suggestions. I revised the summary by removing some of the extraneous details. I hope you find that the revised version is more concise.

---

## [Decision Letter · Decision Letter 2]

16 Jan 2023

PONE-D-22-19509R2In vitro spermatogenesis in isolated seminiferous tubules of immature micePLOS ONE

Dear Dr. Ogawa,

Thank you for submitting your manuscript to PLOS ONE. After careful consideration, we feel that it has merit but does not fully meet PLOS ONE’s publication criteria as it currently stands. Therefore, we invite you to submit a revised version of the manuscript that addresses the points raised during the review process. A major concern that needs to be addressed is to conduct statistical analyses using similar sample size and time points.

We look forward to receiving your revised manuscript.

Kind regards,

Suresh Yenugu

Academic Editor

PLOS ONE

Reviewers' comments:

Reviewer's Responses to Questions

**Comments to the Author**

1. If the authors have adequately addressed your comments raised in a previous round of review and you feel that this manuscript is now acceptable for publication, you may indicate that here to bypass the “Comments to the Author” section, enter your conflict of interest statement in the “Confidential to Editor” section, and submit your "Accept" recommendation.

Reviewer #2: (No Response)

Reviewer #3: All comments have been addressed

2. Is the manuscript technically sound, and do the data support the conclusions?

Reviewer #2: Partly

Reviewer #3: Yes

3. Has the statistical analysis been performed appropriately and rigorously? 

Reviewer #2: No

Reviewer #3: Yes

4. Have the authors made all data underlying the findings in their manuscript fully available?

Reviewer #2: Yes

Reviewer #3: Yes

5. Is the manuscript presented in an intelligible fashion and written in standard English?

Reviewer #2: No

Reviewer #3: Yes

6. Review Comments to the Author

Reviewer #2: It has been improved to some extent by reorganization of the title, figures and figure legends, and additional information of sample size and oxygen level. On the other hand, the reliability of data in terms of time points and sample size are still not clear.

1) In figure 4B, 20% (n= 36), 15% (n=41) and 10% (n=27) oxygen level is compared in terms of the GFP721 and mCherry positivity of seminiferous tubules. The big difference in sample size between the groups gives rise to thought about potential of bias, and it may cause some possible errors in statistical analyses.

2) In figure 4B, the authors stated that “GFP721 positivity was judged during culture days 20 to 30. The mCherry expression was confirmed on culture day 30 or 36.” Both 6 and 10-day difference in in vitro culture of seminiferous tubules is still excessive in order to collect a reliable data from the experiments for two reasons:

1- The duration of mouse spermatogenesis is approximately 34 days. In Acr-GFP/Histone H3.3.3-mCherry double (Acr/H3) transgenic mice, GFP expression starts from pachytene spermatocytes at stage 4 and mCherry is expressed in spermatids from step 11 onward. Since the content and the size of subgroups of germ cells changes within the 6 and 10-day time interval, it is not suitable to include the data collected from day 20 to 30, and 30 to 36.

2-This time gap may affect the viability of germ cells, and number of labelled cells in the meantime.

The authors are recommended to reperform the statistical analysis by using same time points and sample sizes in the groups.

Reviewer #3: (No Response)

7. PLOS authors have the option to publish the peer review history of their article (what does this mean?). If published, this will include your full peer review and any attached files.

Reviewer #2: No

Reviewer #3: No

---

## [Author Response · Author response to Decision Letter 2]

7 Feb 2023

I appreciate the reviewer 2 for their kind comments on our manuscript.

In the 'response to the reviewer' file, I made point-by-point replies to the comments and requests from the reviewer. 

Reviewer #2: It has been improved to some extent by reorganization of the title, figures and figure legends, and additional information of sample size and oxygen level. On the other hand, the reliability of data in terms of time points and sample size are still not clear.

1) In figure 4B, 20% (n= 36), 15% (n=41) and 10% (n=27) oxygen level is compared in terms of the GFP721 and mCherry positivity of seminiferous tubules. The big difference in sample size between the groups gives rise to thought about potential of bias, and it may cause some possible errors in statistical analyses.

Response: I appreciate your insightful comments, which prompted us to review the data and analysis in Figure 4B. The number of samples varied from group to group mainly due to the technical difficulty of preparing uncut isolated STs. Even though we prepared comparable numbers of samples in each group at the start of the experiment, some samples were lost prior to evaluation due to fragmentation or slipping of the ST apart from the frame of the PDMS pillars. Nonetheless, we considered that the number of samples and quantity of data were sufficient for a robust statistical analysis. In the process of reviewing the statistical analysis, we did find a careless mistake. In the data for mCherry, the p-value for the comparison of 20% versus 10% O2 should have been marked with a single asterisk (*) rather than a double asterisk (**). This was corrected. 

2) In figure 4B, the authors stated that “GFP721 positivity was judged during culture days 20 to 30. The mCherry expression was confirmed on culture day 30 or 36.” Both 6 and 10-day difference in in vitro culture of seminiferous tubules is still excessive in order to collect a reliable data from the experiments for two reasons:

Response: The Acr-GFP expression was observed beginning as early as culture day 14, which corresponded to a mouse age of 17 dpp when a 3 dpp mouse testis was used, as in Fig. 2. We observed samples every 7 days and recorded the GFP positivity. By judging GFP over culture days 20 to 30, we were therefore provided two chances to observe GFP. If we had narrowed the period or selected one particular date for judgement, we could have misjudged samples as negative even though they expressed GFP on neighboring days. The data in Figure 2B show that such a risk was not merely hypothetical. That is, the ST in Fig. 2B showed GFP expression on days 24 and 31, but no GFP expression on day 28. As described in the manuscript, such fluctuation/oscillation in GFP expression can occur, and we regarded samples such as this as GFP-positive. This was our rationale for the choice of culture days 20 to 30 as judging period.

In the case of mCherry, magnification using an inverted microscope was required in order to faithfully detect the red fluorescence. Therefore, the time period for mCherry detection was at the end of each culture experiment, and we removed the tissue from the culture dish and placed it directly on a slide glass for observation. As mentioned above, in vitro spermatogenesis does not proceed as smoothly as that in vivo, and in our experience the appearance of mCherry is delayed for days or more. Thus, in the initial experiment, we chose culture day 30, which corresponds to a mouse age of 36 days, as the time point for judging mCherry positivity, because the testis-donor mouse was 6 days old. We observed one mCherry-positive ST in the 10% O2 group, and no mCherry-positive STs in the 15% and 20% groups. In later experiments using 5-day-old mice, we chose culture day 36, corresponding to a mouse age of 41 days, reasoning that that time point would be more appropriate for finding mCherry-positive spermatids. We found 4 mCherry-positive STs in the 10% O2 group and none in the other groups. In hindsight, therefore, culture day 36 was more appropriate for finding mCherry-positive spermatids than culture day 30. We agree with you that it would have been better to pinpoint a particular day for analysis when trying to detect mCherry, since we had to select a single day in any case. Nonetheless, culture days 30 and 36 were within the appropriate time frame for detecting mCherry fluorescence and yielded reliable data showing the superiority of 10% O2. 

1- The duration of mouse spermatogenesis is approximately 34 days. In Acr-GFP/Histone H3.3.3-mCherry double (Acr/H3) transgenic mice, GFP expression starts from pachytene spermatocytes at stage 4 and mCherry is expressed in spermatids from step 11 onward. Since the content and the size of subgroups of germ cells changes within the 6 and 10-day time interval, it is not suitable to include the data collected from day 20 to 30, and 30 to 36.

Response: In our culture experiment, the progression of spermatogenesis was not as smooth as that in vivo. The first appearance of Acr-GFP should have been around culture day 15 when a 0–1 day neonate was used, but it could have been delayed several days. In addition, even if GFP fluorescence appeared once, it could have disappeared and then re-appeared as shown in Fig. 2B. Therefore, we thought we should take steps to ensure that we did not miss such GFP-positivity, even if was temporary. This was our rationale for choosing a 10-day time span as an observation period for judging GFP fluorescence. 

For mCherry, as mentioned above, the determination can be made only once per experiment. Therefore, it was necessary to select one particular day for the analysis. The selection of this day is a difficult decision. If we had chosen a day too early for mCherry detection, i.e., a day when the most advanced germ cells in ST are round spermatozoa such as those in step 8, mCherry would not be detected in any sample at any oxygen concentration. On the other hand, if the day had been too late for mCherry detection, i.e., a day when step-11 spermatids had already appeared in the ST some time ago, the mCherry-positive spermatids would have degenerated and disappeared in the interim, and we would have lost the opportunity to detect them. In the present experiment, these concerns were realistic, because mCherry-positive spermatids appeared rarely and only in low numbers. In careful consideration of all the above, we chose days 30 and 36, which corresponded to mouse ages of 36 and 41 days, as the time points for observation. In both cases, we observed mCherry-positive STs, supporting the appropriateness of our choice of days for the analysis. Therefore, we think our choice was appropriate and the data showing the superiority of 10% O2 were reliable. 

2-This time gap may affect the viability of germ cells, and number of labelled cells in the meantime.

Response: Time gaps of 6 or 10 days could indeed provide an opportunity for the germ cells to die and disappear, as you indicated. However, they would also provide a chance for reappearance. Such appearances and disappearances of germ cells at particular stages of differentiation were observed as fluctuations in GFP fluorescence in each ST during the culture experiment, and the pattern of these changes was specific to each ST. Therefore, it would be ideal to collect data for a long enough period, whether for GFP or mCherry detection, and then to integrate such data to make a proper determination. In the case of GFP expression, such an approach can be achieved by observing the cells with a stereomicroscope while maintaining the culture. Therefore, we chose 20 to 30 days as the period for GFP evaluation. 

The authors are recommended to reperform the statistical analysis by using same time points and sample sizes in the groups.

Response: For the reasons described above, we chose 20 to 30 days as the period for the judgement of Acr-GFP. In the case of mCherry, culture days 30 and 36 were considered preferable for the reasons mentioned above. Finally, we should mention that we corrected a careless mistake found in Figure 4B (the p-value for the 20% versus 10% O2 comparison as for mCherry should have been marked with a single asterisk (*) rather than a double asterisk (**)). We appreciate your careful reading and kind comments on our manuscript.

---

## [Decision Letter · Decision Letter 3]

21 Feb 2023

PONE-D-22-19509R3In vitro spermatogenesis in isolated seminiferous tubules of immature micePLOS ONE

Dear Dr. Ogawa,

Thank you for submitting your manuscript to PLOS ONE. After careful consideration, we feel that it has merit but does not fully meet PLOS ONE’s publication criteria as it currently stands. Therefore, we invite you to submit a revised version of the manuscript that addresses the points raised during the review process. Minor changes in the conclusion section shall be made as per the suggestion of the reviewer.

We look forward to receiving your revised manuscript.

Kind regards,

Suresh Yenugu

Academic Editor

PLOS ONE

Journal Requirements:

Reviewers' comments:

Reviewer's Responses to Questions

**Comments to the Author**

1. If the authors have adequately addressed your comments raised in a previous round of review and you feel that this manuscript is now acceptable for publication, you may indicate that here to bypass the “Comments to the Author” section, enter your conflict of interest statement in the “Confidential to Editor” section, and submit your "Accept" recommendation.

Reviewer #2: All comments have been addressed

2. Is the manuscript technically sound, and do the data support the conclusions?

Reviewer #2: Yes

3. Has the statistical analysis been performed appropriately and rigorously? 

Reviewer #2: Yes

4. Have the authors made all data underlying the findings in their manuscript fully available?

Reviewer #2: No

5. Is the manuscript presented in an intelligible fashion and written in standard English?

Reviewer #2: Yes

6. Review Comments to the Author

Reviewer #2: The authors provided a good explanation for concerns about the validity and reliability of the data due to differences in sample size and time points of the groups compared. On the other hand, those explanations should be clearly stated in the manuscript in order to increase the readability of the publication.

The authors corrected the error made in the statistical analysis and explained the reasons for changes in time interval and sample size used to evaluate the GFP and mCherry expression reliably and rationally. When the explanations are added to the limitation section, the article will be ready for publication in order to be clearly understood by the readers.

7. PLOS authors have the option to publish the peer review history of their article (what does this mean?). If published, this will include your full peer review and any attached files.

Reviewer #2: No

---

## [Author Response · Author response to Decision Letter 3]

10 Mar 2023

Editor

Thank you for submitting your manuscript to PLOS ONE. After careful consideration, we feel that it has merit but does not fully meet PLOS ONE’s publication criteria as it currently stands. Therefore, we invite you to submit a revised version of the manuscript that addresses the points raised during the review process.

Minor changes in the conclusion section shall be made as per the suggestion of the reviewer.

Response: According to the suggestions from Reviewer #2, we added a phrase, technical difficulty in handling STs, in the conclusion section as one of the limitations of our technique. 

Reviewer #2

Comment 1: The authors provided a good explanation for concerns about the validity and reliability of the data due to differences in sample size and time points of the groups compared. On the other hand, those explanations should be clearly stated in the manuscript in order to increase the readability of the publication.

Response: Thank you for this important suggestion. We added the following passages to the Observations section of the Materials and methods: 

Cultured tissues and ST segments were observed at least once a week under a stereomicroscope equipped with an excitation light for GFP (LeicaM205 FA; Leica, Germany). Acr-GFP begins to be expressed in mice at around 15 dpp in vivo, but its expression could be delayed by several days in vitro. In addition, GFP emission can fluctuate at intervals of several days. Therefore, to avoid false-negative results, GFP positivity was not necessarily determined on a single day; in most cases, two observations with a 7-day interval were taken, resulting in an observation period of greater than 7 days. Samples showing GFP expression during those periods were considered GFP-positive. The GFP-positive portions in each stretch of ST were measured by visual approximation as 0%, 1–10%, 11–20%, 21–40%, 41–60%, 61–80% or 81–100%.

 H3.3 mCherry appears beginning at around 28 dpp in vivo. In vitro, however, its expression is delayed days to weeks, depending on the sample. For the reliable identification of mCherry, each sample tissue was removed from the culture well, placed on a slide glass and observed with an inverted microscope (IX73; Olympus) or a confocal microscope (FV1000–MPE; Olympus). The observation timing was carefully determined in each case, taking the status of preceding GFP expression into account. 

Comment 2: The authors corrected the error made in the statistical analysis and explained the reasons for changes in time interval and sample size used to evaluate the GFP and mCherry expression reliably and rationally. When the explanations are added to the limitation section, the article will be ready for publication in order to be clearly understood by the readers.

Response: In the conclusion, we added “technical difficulty in handling STs” as one of the limitations of this technique. In the Results section, we also added the following sentences to clarify the reason for the different numbers of samples among groups. 

The number of samples varied among the three groups due to technical difficulties in the preparation of uncut-isolated STs. Namely, some samples were lost prior to evaluation due to fragmentation or slipping of the ST apart from the frame of the PDMS pillars. Nonetheless, this experiment clearly demonstrated that lower O2 concentrations were associated with a higher rate of GFP expression in a larger area.

We appreciate your comments. We believe the manuscript was substantially improved through your input.

---

## [Decision Letter · Decision Letter 4]

16 Mar 2023

In vitro spermatogenesis in isolated seminiferous tubules of immature mice

PONE-D-22-19509R4

Dear Dr. Ogawa,

We’re pleased to inform you that your manuscript has been judged scientifically suitable for publication and will be formally accepted for publication once it meets all outstanding technical requirements.

Kind regards,

Suresh Yenugu

Academic Editor

PLOS ONE

Additional Editor Comments (optional):

Reviewers' comments:

Reviewer's Responses to Questions

**Comments to the Author**

1. If the authors have adequately addressed your comments raised in a previous round of review and you feel that this manuscript is now acceptable for publication, you may indicate that here to bypass the “Comments to the Author” section, enter your conflict of interest statement in the “Confidential to Editor” section, and submit your "Accept" recommendation.

Reviewer #2: All comments have been addressed

2. Is the manuscript technically sound, and do the data support the conclusions?

Reviewer #2: Yes

3. Has the statistical analysis been performed appropriately and rigorously? 

Reviewer #2: Yes

4. Have the authors made all data underlying the findings in their manuscript fully available?

Reviewer #2: Yes

5. Is the manuscript presented in an intelligible fashion and written in standard English?

Reviewer #2: Yes

6. Review Comments to the Author

Reviewer #2: All of the comments have been addressed efficiently by the authors. The the manuscript is suitable for acceptance of publication.

7. PLOS authors have the option to publish the peer review history of their article (what does this mean?). If published, this will include your full peer review and any attached files.

Reviewer #2: No

---

## [Editor Report · Acceptance letter]

28 Mar 2023

PONE-D-22-19509R4 

*In vitro* spermatogenesis in isolated seminiferous tubules of immature mice 

Dear Dr. Ogawa:

I'm pleased to inform you that your manuscript has been deemed suitable for publication in PLOS ONE. Congratulations! Your manuscript is now with our production department. 

Kind regards, 

on behalf of

Dr. Suresh Yenugu 

Academic Editor

PLOS ONE